# R-locus for roaned coat is associated with a tandem duplication in an intronic region of *USH2A* in dogs and also contributes to Dalmatian spotting

**Takeshi Kawakami**[1]*, **Meghan K. Jensen**[1], **Andrea Slavney**[1], **Petra E. Deane**[1¤a], **Ausra Milano**[1], **Vandana Raghavan**[1], **Brett Ford**[1], **Erin T. Chu**[1¤b], **Aaron J. Sams**[1], **Adam R. Boyko**[1,2]*

**1** Embark Veterinary, Inc., Boston, Massachusetts, United States of America, **2** Department of Biomedical Sciences, College of Veterinary Medicine, Cornell University, Ithaca, New York, United States of America

¤a Current address: Mascoma LLC Lallemand Corporation, Libanon, NH, United States of America
¤b Current address: Amazon Web Services, Inc, Seattle, Washington, United States of America
* adam@embarkvet.com (ARB); tkawakami@embarkvet.com (TK)

**Data Availability Statement:** Our raw data deposited in Dryad is now published and available

## Abstract

Structural variations (SVs) represent a large fraction of all genetic diversity, but how this genetic diversity is translated into phenotypic and organismal diversity is unclear. Explosive diversification of dog coat color and patterns after domestication can provide a unique opportunity to explore this question; however, the major obstacle is to efficiently collect a sufficient number of individuals with known phenotypes and genotypes of hundreds of thousands of markers. Using customer-provided information about coat color and patterns of dogs tested on a commercial canine genotyping platform, we identified a genomic region on chromosome 38 that is strongly associated with a mottled coat pattern (roaning) by genome-wide association study. We identified a putative causal variant in this region, an 11-kb tandem duplication (11,131,835–11,143,237) characterized by sequence read coverage and discordant reads of whole-genome sequence data, microarray probe intensity data, and a duplication-specific PCR assay. The tandem duplication is in an intronic region of usherin gene (*USH2A*), which was perfectly associated with roaning but absent in non-roaned dogs. We detected strong selection signals in this region characterized by reduced nucleotide diversity ($\pi$), increased runs of homozygosity, and extended haplotype homozygosity in Wirehaired Pointing Griffons and Australian Cattle Dogs (typically roaned breeds), as well as elevated genetic difference ($F_{ST}$) between Wirehaired Pointing Griffon (roaned) and Labrador Retriever (non-roaned). Surprisingly, all Dalmatians (N = 262) carried the duplication embedded in identical or similar haplotypes with roaned dogs, indicating this region as a shared target of selection during the breed's formation. We propose that the Dalmatian's unique spots were a derived coat pattern by establishing a novel epistatic interaction between roaning "R-locus" on chromosome 38 and an uncharacterized modifier locus. These results highlight the utility of consumer-oriented genotype and phenotype data in the discovery of genomic regions contributing to phenotypic diversity in dogs.

with the provided DOI (doi:10.5061/dryad.qz612jmdt) or URL (https://doi.org/10.5061/dryad.qz612jmdt).

**Funding:** This study was funded by Embark Veterinary, Inc. and the participants that provided DNA and phenotypic information via Embark's web-based platform. The funder only provided financial support in the form of salaries for all authors but did not have any additional role in the study design, data collection and analysis, decision to publish, or preparation of the manuscript.

**Competing interests:** I have read the journal's policy and the authors of this manuscript have the following competing interests: TK, MKJ, AS, AM, VR, BF, AJS and ARB are employees of Embark Veterinary, a canine DNA testing company which will offer commercial testing for the variant described in this study. ARB is co-founder and part owner of Embark. PED and ETC were employees of Embark Veterinary when this study was conducted but were employees of Mascoma LLC Lallemand Corporation and Amazon Web Services, respectively by the time of manuscript submission. Mascoma LLC Lallemand Corporation and Amazon Web Services do not have any competing interests with this study and do not alter our adherence to PLOS ONE policies on sharing data and materials.

## Introduction

Diverse coat colors and patterns observed in dogs and other domestic animals have long fascinated breeders. Geneticists and evolutionary biologists have taken interest in this remarkable diversity because it provides an excellent model to study 1) how numerous phenotypes can be established from morphologically less diverse wild counterparts, 2) how changes in selection pressure alter genotype-phenotype interactions, and 3) how repeated evolution of similar coat phenotypes in multiple species is attainable. Years of research on the genetics of coat color have led to the identification of a number of genes that have been repeatedly involved in the formation of diverse color morphs in both wild and domesticated animals [1]. One of the well-known genes involved in the repeated evolution of light and dark color polymorphisms is the *melanocortin-1 receptor* (*MC1R*) that modulates the activity of the melanin synthesis pathways in the fur, plumage, scales and skin of mammals, birds, reptiles, amphibians, and fish [2]. In other cases, similar coloration has independently evolved in multiple lineages via mutations in different genes (e.g., *LYST* and *AIM1* in polar bears and *KIT* and *MATP* in horses with white coats) [3–5]. Understanding the genetic mechanisms of color variation and phenotypic convergence has shed light on how novel phenotypes evolve under similar selective forces (either natural or artificial). In addition, high conservation of melanogenesis pathways across vertebrates warrants transformative research in human genomics research, such as the case of *MC1R* that is strongly associated with increased risk for melanoma [6,7].

Ticking and roaning are two common coat patterns observed in dogs and other domestic animals. Ticking is characterized as small pigmented spots of varying numbers and sizes appearing on otherwise unpigmented (white) areas. Roaning is similar to and sometimes co-occurs with ticking but consists of pigmented and unpigmented hairs interspersed more evenly without the formation of distinct spots. The distinctive spots of Dalmatians have been believed to be a modified form of ticking where a size of each tick or spot is enlarged and distinctive by a modifier locus (flecking locus, F-locus) mapped on canine chromosome (CFA) 3 [8]. Typically, individuals are born without ticking, Dalmatian's spotting, and roaning patterns, but instead these pigmented areas develop as the individual ages, indicating time-dependent action of underlying pigmentation mechanisms. In dogs, two theoretical loci, namely "T-locus" and "R-locus", have been proposed as responsible loci for ticking and roaning, respectively, although it is unclear whether they constitute the same genomic regions with different variants because they have not been characterized at a molecular level [9]. Similarly, the molecular basis of F-locus is unknown. *KIT ligand gene* (*KITLG*) has been reported as a causal gene for roaning in cattle [10,11], pigs [12,13], and goats [14], but does not seem to be involved in roaning in dogs [9].

Gene interaction or epistasis is one of the key mechanisms in the formation of phenotypic diversity in both wild and domesticated species. A well-known example is three color types of Labrador Retrievers, where *tyrosinase-related protein 1* (*TYRP1*) and *MC1R* determine their coat colors as black, chocolate, or yellow [15]. Modifier genes constitute a type of epistasis; for example, several variants of *microphthalmia-associated transcription factor* (*MITF*) modify the coat color of dogs by preventing the melanocyte development and migration in certain areas of the body and, in some cases, across nearly the entire body. This results in a loss of pigmentation leading to white markings in otherwise uniformly colored areas [16–18]. S-locus is the major locus controlling this white spotting pattern, and several variants within and close to *MITF* have been identified, including a SINE insertion at 3 kilobase (kb) upstream of the *MITF* transcription start site (TSS) and a variable length polymorphism (Lp) at 100 bp upstream of *MITF* TSS [17,18]. Both T-locus and R-locus are considered as modifier loci by locally changing coat color from white to pigmented through the interaction with S-locus [9].

Furthermore, it has been suggested that unique spots found exclusively in Dalmatians are a result of an epistatic interaction between T-locus and F-locus on CFA3 in linkage with a hyper-uricosuria-associated variant in *SLC2A9* [19]; however, it is not known how *SLC2A9* itself or other genes linked to it can modify pigmentation patterns via T-locus.

Here we investigate genomic regions associated with ticking and roaning coat patterns in dogs by using a total of 1,281 purebred dogs for marker discovery ("discovery panel") and 274 mixed breed dogs for marker validation ("validation panel") that were genotyped at our Embark SNP array with 220,484 markers covering all 38 autosomes and chromosome X (S1 File). Dog owners contributed to this study by providing photographs of their dogs, from which we classified their phenotypes as ticked, roaned, or lacking these patterns to identify genomic regions associated with these phenotypes by genome-wide association study (GWAS). For the discovery of the associated genomic regions, we selected a diverse set of purebred dogs in 27 breeds that commonly exhibit white spotting patterns caused by genetic variants at the S-locus. Pigmented "case" dogs were defined on the basis of specific criteria in the extent of ticking and/or the presence of roaning patterns, whereas unpigmented "control" dogs were selected from the same breed cohorts with white spotting patterns but without any evidence of ticking and roaning. Both case and control dogs included in the study, thus, exhibited white spotting patterns. Dogs without white spotting patterns, including those that appear white as a result of dilute pheomelanin and those with a residual white phenotype were excluded from the study to ensure the uniformity of the base coat color. In addition, we took advantage of publicly available whole-genome re-sequencing data to fine-map the location of a putative causal variant associated with roaning. The genotype-phenotype association was robust as phenotypes of mixed breed dogs (an independent validation dataset) were accurately predicted by their genotypes. Finally, phylogenetic analysis of the putative causal variant was performed to examine signatures of positive selection and identify whether any other coat patterns might be associated with the variant.

## Materials and methods

### Phenotype data collection

Owner-submitted photographs were used to evaluate coat patterns of dogs in the Embark Veterinary database where the owner agreed to participate in scientific research. To ensure a high level of confidence in correctly assessing the coat patterns, the following selection criteria were applied based on the photograph and on the dog itself to determine if each individual was a good candidate for the study. Photographs had to be of high quality, in focus, well-lit, and not show evidence of filter use or image-editing. In addition, photographs that included multiple dogs or that depicted a dog very far from the camera were excluded. A reasonable amount or the entirety of the dog's body had to be shown in the photograph, especially areas where white patterns likely governed by S-locus [17,18] were common (e.g., face, chest, front legs, and feet). Dogs with the following three types of coat patterns were removed from the dataset to minimize the bias between case and control groups as well as to remove dogs with white areas in their coat by other pigmentation loci: "residual white" (a small patch of white on the chest), merle by M-locus (M/M and M/m) [20], and light cream coat likely under the influence of E-locus (e/e) and I-locus (i/i) [21]. It should be noted that the selection of dogs was based on phenotypes without referring to genotypes of these loci except E-locus, where e/e homozygotes (SNP marker at CFA5:63694334 in *MC1R*) were excluded from the study. It is common to see a white patch even on the chest of ticked and roan dogs, thus, control dogs especially had to have white areas extending beyond the chest area into the legs and/or face. Dogs without sufficient white spotting patterns could otherwise be mis-phenotyped since ticking and roan are

not visible on a solid pigmented background. Finally, photos of new-born or juvenile dogs were not considered since ticking and roaning patterns develop with age.

If roaning was observed on any part of the body, the dog was scored as roaned. Similarly, dogs were classified as ticked if they had any spots on their body, and the extent of ticking was scored either the scale one (lightly ticked) or two (heavily ticked) (S1 Fig). Because it has been postulated that ticking and roaning may result from a similar genetic mechanism, roaned dogs were never considered as 'not ticked' controls nor were ticked dogs considered 'not roaned' controls. However, dogs could be considered both ticked and roaned if both patterns were clearly visible in the coat. As of September 2019, we identified 1,281 adolescent and adult dogs whose coat pattern could be assumed to be developmentally complete (approximately 6 months or older) (S2 Fig). A total of 27 breeds were included in the discovery panel (S1 Table). Australian Cattle Dogs, Australian Shepherds, and Border Collies were considered as a herding group, and the remaining 24 breeds were grouped as non-herding dogs. Breeds that never or rarely exhibit white spotting patterns were not included in the discovery panel. These dogs were used for the following association studies to identify markers associated with coat color patterns (i.e., discovery panel dogs). A random subset of 100 dogs were phenotyped by the same individual a second time several weeks later to evaluate phenotyping consistency. Overall concordance was 98% for the roan phenotype (binary) and 98% for the ticking phenotype (no ticking, light ticking, or heavy ticking). All discordant calls were either the changes from the absence of roaning to inconclusive or from heavy to light ticking.

The definition of the F-locus for Flecking is unclear. While the term flecking has been sometimes used synonymously with ticking and/or roaning (e.g., https://images.akc.org/pdf/judges/groups/Sporting_Group.pdf), other definitions suggest that flecking is unpigmented hairs within pigmented areas (i.e., white hairs within a pigmented spot). It has been postulated that one of the distinctions between Dalmatian spotting and ticking is the presence or absence of white hairs within the spots; Dalmatian spots have completely solid pigment while ticks contain interspersed white hairs [22]. We adopted the latter definition and used F-locus as a Dalmatian's spot modifier where a homozygous recessive genotype (*f/f*) makes completely solid spots.

As a separate dataset, we collected coat phenotype data of mixed breed dogs by applying the same phenotype selection criteria with the discovery panel to validate the prediction of coat phenotype based on genetic markers ("validation panel"). To represent a mixed breed dog population, we initially randomly selected 400 dogs with at least one copy of the duplication-associated haplotypes and another 400 dogs without it. Then we applied the same photo selection criteria with the discovery panel to choose 274 mixed breed dogs. To reduce observer bias, all dog's phenotypes in both discovery and validation panels were scored by one individual, MJ, who was blind of the dog genotypes and their genetic ancestry at the time of phenotyping.

## Genotyping and genome-wide association

DNA was extracted from buccal swab samples collected by dog owners and extracted by Illumina, Inc. Genotypes of the dogs were collected by using custom Illumina Canine high-density SNP arrays (a total of 220,484 markers) (S1 File). Mean genotyping rate was 97.4% across all dogs. After removing markers with minor allele frequency less than 1%, we used 176,910 markers, for which the genotyping rate was 99.8%. Genotyping rate calculation and marker filtering were performed by PLINK v1.9 [23].

To identify genomic regions associated with coat color variation, we applied a univariate linear mixed model implemented in GEMMA v0.98.1 [24]. To account for confounding effects of shared ancestry among dogs of the same or closely related breeds, a relatedness matrix was constructed from the genotypes of all autosomal markers and used as a random effect in the model.

We performed case-control GWAS by using dogs with no ticking and roaning as "controls" and either ticked or roaned dogs as "cases". We used the Wald test to detect significant association between markers and phenotype by applying a threshold of $P < 5.0 \times 10^{-8}$. To delineate a region associated with roaning on CFA38, haplotypes of roaned and non-roaned dogs were reconstructed from the array genotypes by using Beagle v4.1 with default parameter settings [25]. We derived genetic map positions from a LD-based canine recombination map [26].

## Identification of tandem duplication

We retrieved LRR (probe intensity) data of 1,182 dogs (346 roaned and 567 control dogs in the discovery dataset, 118 and 151 roaned and control dogs in the validation dataset) using Illumina GenomeStudio v2.0.4 (Illumina Inc., San Diego) to identify regions with putative non-balanced SVs near the markers associated with coat pigmentation patterns. To identify SVs in these regions, we used Manta [27], which uses paired and split-read evidence for SV detection in mapped sequencing reads. To generate mapped sequence reads, we downloaded whole-genome sequence data for 38 dogs of the eight breeds from the NCBI Sequence Read Archive (https://www.ncbi.nlm.nih.gov/sra) (S2 Table). We selected these eight because of the high prevalence of ticking and roaning patterns in these breeds. Sequence reads of these samples were mapped to CanFam3.1 reference genome by using the BWA-MEM algorithm in BWA [28]. We calculated read depths for all sites using the GATK DepthOfCoverage tool [29]. To visualize the CFA38 duplication breakpoints, we calculated mean per site read depths for non-overlapping 5-kb windows along CFA38 and then were divided by the autosome average read depth for normalization. We visualized discordant read pairs with Integrative Genomics Viewer (IGV) [30]. To identify haplotypes associated with the CFA38 duplication, we phased genotypes of bi-allelic variants of 722 dogs and other canid species [31] with Beagle v4.1 with default parameter settings [25].

We validated haplotypes associated with the CFA38 duplication by a breakpoint PCR assay. We designed three pairs of primers to amplify three regions in separate PCR reactions: 1) the midpoint spanning the duplication (midpoint primer pair), 2) 5' flanking region of the duplication start region (5' control primer pair), and 3) 3' flanking region of the duplication end region (3' control primer pair) (S3 Table). One microliter of total DNA was used for PCR reactions using the following primer combinations: Tick38-F2-2 and Tick38_R1 (midpoint primer pair), Tick38_F1 and Tick38_R1 (5' control primer pair), and Tick38-F2-2 and Tick38-R2-2 (3' control primer pair). All PCR reactions were performed using Go Taq G2 Hot Start Green Master Mix (Promega M7422) in a total volume of 25 uL following the manufacturer's protocol. The following cycling parameters were used: 95°C 3min, 35X (95°C 30s, 58°C 30s, 72°C 30s), 72°C 5m, 12°C hold. PCR product was visualized on a 1.5% agarose gel with 1X GelRed (Biotium Cat No 41003); the products from three dogs were submitted for purification and Sanger sequencing at Biotechnology Resource Center at Cornell University.

## Detecting signatures of selection

Pairwise nucleotide diversity ($\pi$) was calculated using VCFTools v0.1.16 [32] for Wirehaired Pointing Griffons, Border Collies, and Labrador Retrievers, separately in 500-kb sliding windows with 10-kb steps along CFA38. Genetic differentiation was measured as $F_{ST}$ between breeds (Wirehaired Pointing Griffon vs. Border Collies and Labrador Retrievers vs. Border Collies) in the same window sizes. Whole-genome variant data reported in ref [31] were used. Sites with missing genotype rates larger than 50% were excluded. Next, we used the Embark array genotype data of the discovery panel to calculate runs of homozygosity (ROH) and cross-population extended haplotype homozygosity (XP-EHH) [33,34]. ROH was calculated

for Australian Cattle Dogs, Dalmatians, and Border Collies by using PLINK v1.9 [23] in Australian Cattle Dogs, Dalmatians, Border Collies, and Labrador Retrievers with the following parameters (following ref [35]):

- homozyg-window-het 0

- homozyg-snp 41

- homozyg-window-snp 41

- homozyg-window-missing 0

- homozyg-window-threshold 0.05

- homozyg-kb 500

- homozyg-density 5000 (set high to ignore)

- homozyg-gap 1000 (set high to ignore)

The frequency of ROH at each marker position was calculated by dividing the sum of ROH state (absence or presence as 0 or 1, respectively) by the total number of individuals. This indicated the proportion of autozygous individuals at a given marker position along a chromosome. XP-EHH was calculated for Australian Cattle Dogs, Dalmatians, and Labrador Retrievers (with Border Collies as a reference breed) by using rehh R package [34].

### Ethics statement

Participating dogs were part of the Embark Veterinary, Inc. customer base. Owners provided informed consent to use their dogs' data in scientific research by agreeing the following statement: "I want this dog's data to contribute to medical and scientific research". Ethical approval was not required as non-invasive methods for genotype or phenotype collection were used (buccal swab and photographing, respectively). Dogs were never handled directly by researchers. Owners were given the opportunity to opt-out of the study at any time during data collection. The discovery and validation cohorts were selected from data available before February 2020. All published data have been de-identified of all Personal Information as detailed in Embark's privacy policy (embarkvet.com/privacy-policy/).

## Results

### A novel association on chromosome 38 with roaning (but not with ticking)

We selected a total of 1,281 purebred dogs with profile pictures where dogs showed white spotting patterns in their bodies (S1 Table). Inspection of customer-provided photographs identified 344 dogs with varying degrees of ticking, 358 dogs with a roaning pattern on some part of the body, and 579 dogs without any noticeable ticking or roaning in any part of their bodies (i.e., "control" dogs). Dogs that exhibited both phenotypes (ticking and roaning) were excluded from the study. The extent of ticking was further classified as either lightly ticked (n = 168) or heavily ticked (n = 176), and these ticking scores were treated as an ordinal variable in the subsequent GWAS (together with the control dogs). Since dogs with residual white and dilute pheomelanin were excluded from the study (see Methods), we assumed that all dogs selected for the study with white areas had one or two copies of the following S-locus alleles: extreme white ($s^w$), piebald ($s^p$), and irish white ($s^i$) [17,18].

Our GWAS of 358 roaned dogs (cases) and 579 non-ticked, non-roaned dogs (controls) identified three significant markers (Figs 1A and S3). The most significant marker was at the

position 11,085,443 on CFA38, an exonic region of usherin gene (*USH2A*) ($P$ = 6.9 x $10^{-46}$). The second most significant marker was on CFA13 at the position 8,625,896 that was in an intronic region of R-spondin 2 gene (*RSPO2*) ($P$ = 1.1 x $10^{-29}$). The association with *RSPO2* likely resulted from the non-roaned and roaned breeds also commonly having contrasting coat texture (e.g., English Springer Spaniels and German Wirehaired Pointers, the latter having a wiry coat, known as "furnishing") [36]. The marker at 8,625,896 on CFA13 was 15-kb away from the most likely causal variant of furnishing (167-bp insertion within the 3′UTR of *RSPO2* at position 8,610,419). Similarly, the association with the marker on CFA32 at the position 4,509,367 ($P$ = 6.8 x $10^{-17}$) was probably due to the difference in fur length between the case and control groups, such as Border Collies (long-haired usually without roaning) and Australian Cattle Dogs (short-haired with roaning), because this marker, lying in exon 1 of fibroblast growth factor 5 (*FGF5*), was indeed known to be the most likely causal variant of fur length [36]. To minimize the effect of genes associated with breed-specific characteristics, we re-ran additional GWAS, subdividing the dataset by herding breeds (Australian Cattle Dogs, Australian Shepherds, and Border Collies) and the rest of breeds (hereafter referred to as non-herding breeds). As expected, the GWAS association signals on CFA13 and CFA32 were not consistently detected in GWAS for herding (N = 268 and 495 for roaned and control dogs, respectively) and non-herding breeds (N = 90 and 84 for roaned and control dogs, respectively) (S4 Fig). On the other hand, the marker on CFA38 remained highly significant in both herding and non-herding groups ($P$ = 1.5 x $10^{-27}$ and $P$ = 3.3 x $10^{-17}$, respectively). A weakly associated marker was found at the position 21,848,176 on CFA20, an intron of *MITF* ($P$ = 1.5 x $10^{-7}$) (Fig 1), and the association was more pronounced in GWAS for herding breeds ($P$ = 5.4 x $10^{-19}$) (S4 Fig). This is probably due to the difference in the frequency of S-locus alleles between breeds, where the base coat color of Australian Cattle Dogs is often mostly white with a likely genotype of $s^w$/$s^w$, whereas the piebald and Irish spotting white coat patterns are more common in Australian Shepherds and Border Collies with a likely genotype of $s^p$/- or $s^i$/- [17,18].

In contrast to the GWAS for roaning, the GWAS for the ticking pattern did not reveal strongly associated markers (N = 579, 168, and 176 for controls, lightly ticked, and heavily ticked, respectively) (Figs 1B and S3). The same markers at *MITF* on CFA20 and *USH2A* on CFA38 were detected as marginally significant when both herding and non-herding breeds were analyzed together ($P$ = 3.5 x $10^{-8}$ and $P$ = 3.6 x $10^{-8}$, respectively). However, when herding and non-herding breeds were analyzed separately, only the *MITF* marker was significant in herding breeds (N = 75 and 495 for ticked and control dogs, respectively), whereas no marker was significantly associated with ticking in non-herding breeds (N = 269 and 84 for cases and controls, respectively) (S5 Fig). Similar to roaning, this was likely due to the difference in the frequency of S-locus alleles in herding breeds since Australian Cattle Dogs were mostly ticked with large white areas in our dataset.

## Identification of an 11-kilobase tandem duplication

The association between the marker on CFA38 and roaning coat phenotype was robust regardless of whether herding or non-herding breeds were used, whereas this marker was not strongly associated with ticking. Thus, we focused our analysis on the region surrounding this marker to further characterize the genetic variants associated with roaning. The roan-associated region on CFA38, defined as a region containing markers with $P$ < 5.0 x $10^{-8}$ (CFA38:10,985,456–11,380,922), had a protein-coding gene, *USH2A* (S5A Fig). The non-roaned control group was completely devoid of the roan-associated "A" allele at the most significant marker at the position 11,085,443 on CFA38, while 57% and 38% of roaned dogs were

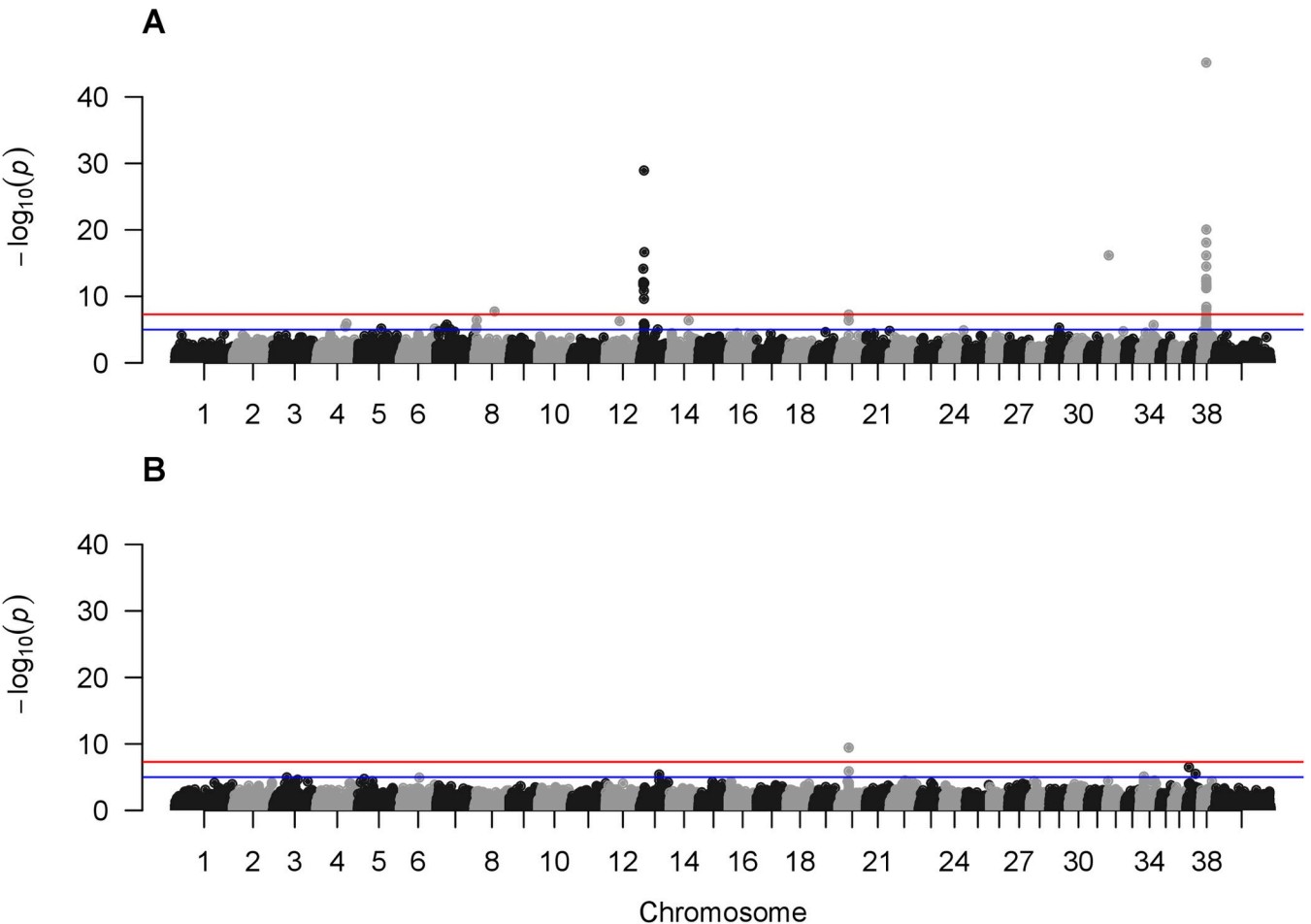

**Fig 1. Manhattan plots of association with roaning and ticking.** A) Roaning. B) Ticking. Red and blue horizontal lines are significant ($P < 5 \times 10^{-8}$) and suggestive ($P < 1 \times 10^{-5}$) associations, respectively.

AA homozygous and AG heterozygous, respectively, indicating a dominant action of this locus. A total of 321 haplotypes were identified in this region based on 52 markers, among which 21 haplotypes had the roan-associated "A" allele at the position 11,085,443 (S5B Fig and S5 Table). There were 16 roaned dogs (4%) without the roan-associated "A" allele (i.e., GG genotype at the position 11,085,443), five of which were homozygous across the 52 markers (three dogs with hap_GG01 homozygotes and two dogs with hap_GG02 homozygotes; S5B Fig). The 21 haplotypes with the "A" allele shared an identical haplotype from the position 11,006,085 to 11,191,833 except for three rare haplotypes (hap_07, hap_12, and hap_21). The region between 11,006,085 and 11,072,648, where hap_07 was different from the rest of the haplotypes, was probably not involved in roaning because there were three control dogs that were homozygous and identical to hap_07 in this region (dog_1746, dog_1758, and dog_1807) (S5 Table). Thus, we narrowed down a roan-associated region to a 119-kb region (11,072,648–11,191,833) spanning 14 exons and 14 introns of *USH2A*.

There were 4,569 previously known single nucleotide variants (SNVs) and small indels in this 119-kb region based on publicly available whole genome resequencing of 722 dogs and canid species (hereafter referred to as WGS data) [31]. Variant Effect Predictor (VEP) [37] identified six high-impact and 61 moderate-impact variants out of these 4,569 variants (S9

File). All 16 dogs of breeds where roaning was common (S2 Table) were wildtype homozygotes at the six high-impact variant sites. There were three moderate-impact variants where frequencies of mutant variants were higher than 50% in the 16 dogs, one of which was the most significant GWAS marker (CFA38:11,085,443) (S8 File). This variant was a missense mutation (G > A) leading to an amino acid change from proline to serine of *USH2A* protein (Fig 2). However, roughly 5% of roaned dogs were GG homozygotes (S6 Table). The remaining two moderate-impact variants were missense mutations: C > T at CFA38:11,111,286 leading to glutamic acid to lysine substitution and G > A at CFA38:11,169,445 leading to proline to serine substitution. To evaluate the association between phenotypes and genotypes of these moderate-to-high impact variants in our sample cohort, we imputed these genotypes by using IMPUTE2 [38] (S1 Text). All 358 roaned dogs had at least one missense allele at these moderate-impact variant sites; however, there were 48 and 136 non-roaned dogs with two copies of the missense allele at these sites, respectively, suggesting that these variants were unlikely to be the causal variant (S6 Table).

Since Variant Effect Predictor (VEP) [37] suggested that none of the moderate-to-high impact variants were perfectly associated with roaning, we next searched for structural variations (SVs) as candidate causal variants for roaning by using probe intensity of the Illumina microarray (log R ratio: LRR). We found that the mean LRR was significantly higher in roaned dogs than control dogs at a marker position 11,140,991 on CFA38, 55-kb away from the most significant GWAS marker (probe intensity = 0.171 and -0.031, N = 346 and 566 for roaned and control dogs, respectively; t-test, t = 29.504, P = 5.2 x $10^{-130}$). Stronger probe intensity indicated the existence of a duplicated copy of this region. There was no difference in the probe intensity at this marker between ticked and control dogs (-0.018 and -0.021, N = 55 and 208 for ticked and control dogs, respectively; t-test, t = -0.858, P = 0.39).

To identify a duplication associated with roaning, we searched for SVs by using the WGS data (S2 Table) [31]. Manta, an SV caller based on short-read sequence data [27], identified an

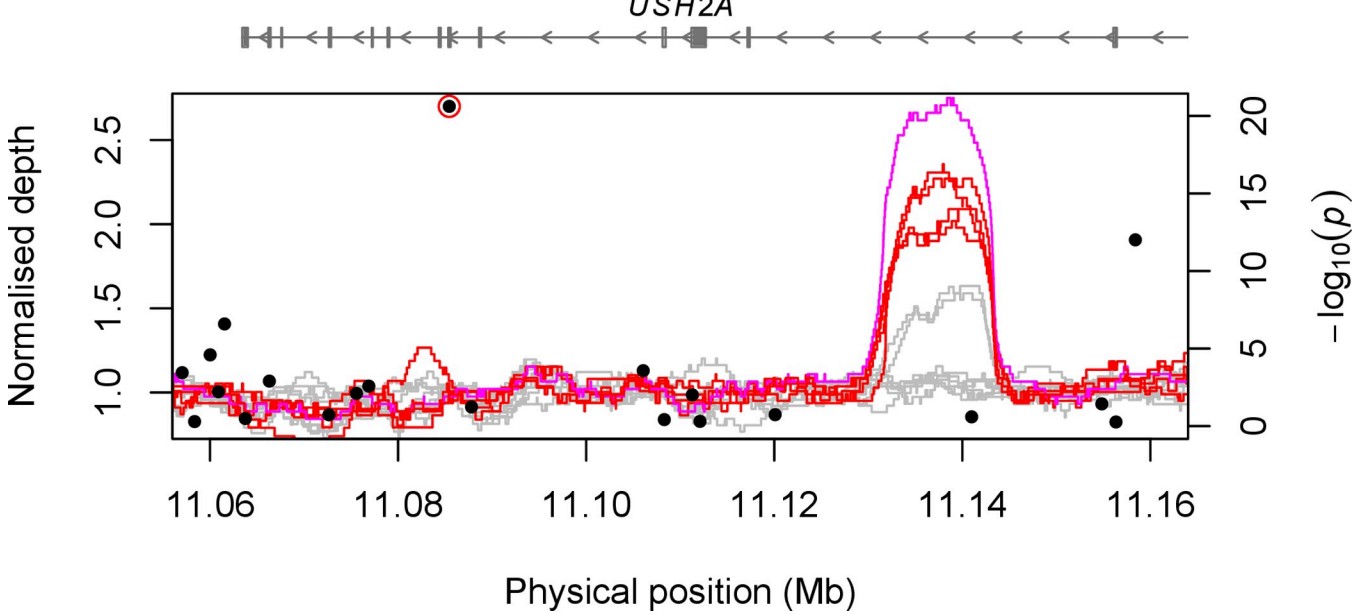

**Fig 2. Normalized read depth in 5-kb sliding windows across the significant GWAS locus on CFA38 for Australian Cattle Dogs (red), German Wirehaired Pointer (pink), and Border Collies (grey).** Filled circles show the corresponding markers of the Manhattan plot shown in Fig 1A (red circle: Most significant marker).

11.4-kb duplication (CFA38:11,131,835–11,143,237) in 15 out of 16 dogs of breeds where roaning was common (4 Australian Cattle Dog, 1 German Wirehaired Pointer, 1 Wirehaired Pointer, and 11 Wirehaired Pointing Griffon). Five dogs in three breeds where ticking is common did not carry the duplication (1 Brittany, 1 Clumber Spaniel, and 3 English Setter), while 3 out of 15 Border Collies carried the duplication (S2 Table). Read depth of the dogs with the duplication was 1.5–2.5 times higher in regions between 11.13 Mb and 11.14 Mb than the genome-wide average (Fig 2). Border Collies did not show such an abrupt increase except for the three dogs where the tandem duplication was identified by Manta. Discordant read pairs facing outwards were observed in dogs with elevated read depth in the duplicated region, suggesting that the duplication was in a tandem orientation (S6 Fig). Out of 15 potentially roaned dogs with the duplication, 10 dogs were homozygous at the most significant GWAS marker on CFA38 and shared a long haplotype 100-kb upstream and downstream of the duplication (hereafter referred to as the duplication-associated haplotype) (Fig 3). The remaining five dogs with the duplication had either one copy of the duplication-associated haplotype or a potential recombinant haplotype of the duplication-associated haplotype by sharing a core haplotype from the positions 11,122,646–11,167,876. The dogs without the duplication did not have the duplication-associated haplotype or similar ones. Interestingly, two Dalmatians in the WGS data were both homozygous for the duplication-associated haplotype (Fig 3).

To test whether the duplication-associated haplotype identified in the WGS data was indeed associated with the tandem duplication in our GWAS discovery panel dogs, we designed a breakpoint PCR assay by targeting the region spanning the two copies (forward and reverse primers mapping to CanFam3.1 CFA38:11,143,136–11,143,155 and CFA38:11,131,969–11,131,988, respectively) (Fig 4 and S3 Table). We assayed a total of 97 dogs (57 and 40 dogs with or without roaning, respectively) (S5 Table). This primer pair produced a single 400-bp amplicon in all roaned dogs with at least one copy of the roan-associated "A" allele at the position 11,085,443 (N = 52). Of the 16 roaned dogs without the roan-associated "A" allele (S5 Table), five samples were available for the breakpoint PCR assay. All of these five samples produced the 400-bp amplicon. There was one homozygous dog and one heterozygous dog for

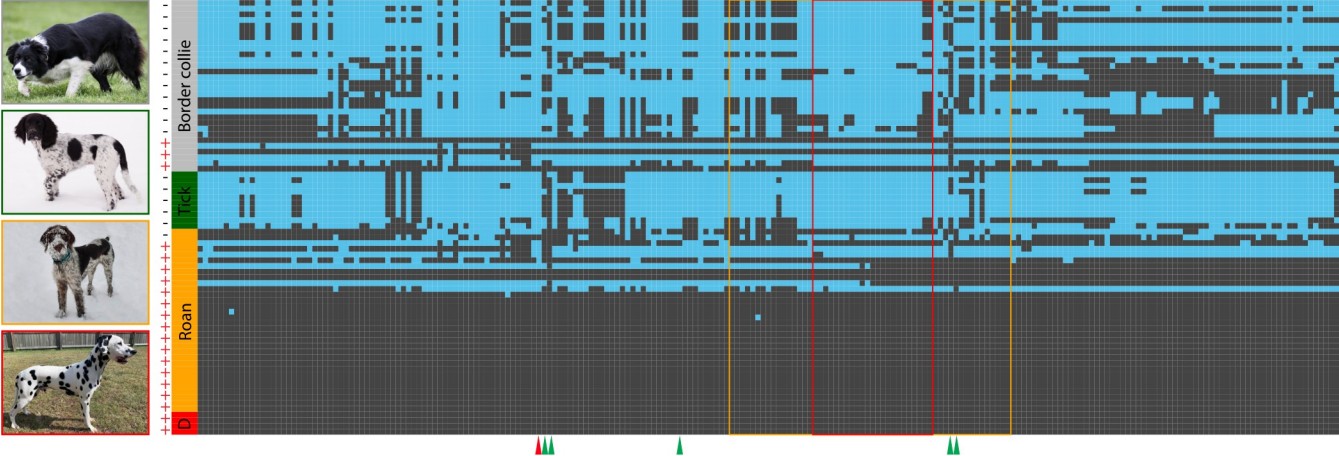

**Fig 3. Haplotypes near the marker on CFA38 significantly associated with roaning.** Border Collies (grey), breeds with high frequency of ticking (green), breeds with high frequency of roaning (orange), and Dalmatians (red). Rows correspond to haplotypes (two rows/individual), and columns correspond to markers. The positions of the first and last markers are 11,031,835 and 11,243,237, respectively. +/-: presence and absence of the 11-kb duplication based on Manta. Red box: 11-kb duplication (CFA38:11,131,835–11,143,234). Yellow box: a core haplotype (CFA38:11,122,646–11,167,876). Red triangle: the most significant marker associated with roaning. Green triangle: markers used for defining the duplication-associated haplotypes. Photos of representative breeds are shown (from top to bottom: Border Collie, Münsterländer, German Wirehaired Pointer, and Dalmatian). Photo credit: Alison Rule, Brett Ford, Dominic Ebacher, Hanna N.

**Fig 4. PCR genotyping of the tandem duplication on CFA38 associated with roaning.** A) Schematic view of the design of the PCR genotyping assay. Yellow boxes indicate the duplicated region. Single-headed arrows indicate pairs of primers to amplify three regions. The first (black) and the third (yellow) primer pairs should produce amplicons in all dogs regardless of the presence or absence of the duplication, while the second pair in the middle should produce an amplicon only in dogs carrying the duplication. Representative coat patterns of non-roaned (top) and roaned dogs (bottom) are shown (left: non-herding group, right: herding group) B) PCR genotyping of a roaned and control dogs. Each gel lane corresponds to PCR primer pairs depicted in panel A. Photo credit: Kellina H., Fernanda Lesnau, Lisa Hayden, and L. Bray.

hap_GG01, indicating a potential recombination event between the markers at 11,120,096 and 11,140,091 (S5B Fig). Two dogs were heterozygous for hap_GG02, which was also likely a recombinant with the duplication because two other dogs homozygous for hap_GG02 were also both roaned. The fifth dog did not carry any of these two haplotypes, but one of the haplotypes, hap_r56, was identical to hap_GG02 from the position 11,076,885 to 11,156,338, indicating the presence of the duplication in this haplotype. Of the remaining 11 roaned dogs without the roan-associated "A" allele, nine dogs had at least one copy of hap_GG01 or hap_GG02, supporting the association between the duplication and roan phenotype. Two dogs without these two haplotypes (dog_1774 and dog_2063) also carried a possible recombinant haplotype: hap_r05 (identical to hap_GG01 from the position 10,985,456 to 11,298,034), and hap_r57 (identical to hap_GG02 from the position 11,072,648 to 11,380,922) (S5 Table). Thus, we assumed that the following haplotypes carried the duplication: all 21 haplotypes with the roan-associated "A" allele (hap_01—hap_21), hap_GG01, hap_GG02, hap_r04, hap_r05, hap_r56, hap_r57 (S5 Table and S7 Fig). All control dogs did not have any of these haplotypes, and none of them tested for the PCR assay produced the 400-bp amplicon (N = 40).

To confirm the presence of the duplication-associated haplotype that was found in two Dalmatians in the WGS data, we used additional data of 262 purebred Dalmatians genotyped by the SNP array (S5 Table). Two haplotypes with the "A" allele at the position 11,085,443 (hap_01 and hap_06) were common, and 98% of Dalmatians (257 dogs) had at least one copy of these haplotypes. The remaining 2% of Dalmatians (N = 5) were homozygous for hap_d003, which was identical to hap_01 from the position 11,112,104 to 11,380,922 (S7 Fig). We confirmed the presence of the duplication in Dalmatians with these haplotypes (hap_01, hap_06, and/or hap_d003) by the breakpoint PCR assay (n = 6) (S5 Table). We, thus, concluded that all 262 Dalmatians carried at least one of these haplotypes with the duplication.

Since the CFA38 duplication was more strongly associated with the roaning, we hypothesized that the duplication itself or neighboring mutations were the causal variant. The CFA38 duplication was in an intronic region of *USH2A*, and the orthologous region in the human reference genome (hg38) was at chr1:215,694,945–215,712,452 based on Liftover [39]. At least three clusters of highly conserved sequences were identified in this region (maximum PhyloP scores of 5.5, 4.3, and 4.1), which overlapped with a DNAse I hypersensitive sites and transcription factor binding sites annotated by Open Regulatory Annotation (ORegAnno) (S9 Fig). In addition, there were two additional regions of high conservation outside the duplication (maximum PhyloP scores of 9.6 and 9.1), which were annotated as transcription factor binding sites by ORegAnno and interaction regions by GeneHancer based on Hi-C mapping.

## Phenotype and genotype association

The presence or absence of the tandem duplication on CFA38 was predicted for the discovery panel dogs based on the haplotypes associated with the duplication (S5 Table). A total of 357 dogs had at least one copy of the duplication-associated haplotypes. The presence of the duplication-associated haplotypes explained all roaned cases (246 homozygous and 112 heterozygous dogs out of 358 roaned dogs), whereas these haplotypes were absent in non-roaned dogs. These duplication-associated haplotypes were able to predict the roaned coat pattern more accurately than the genotypes of the most associated SNP in our GWAS (chi-squared test, $P = 3.4 \times 10^{-204}$ for the duplication and $P = 1.4 \times 10^{-188}$ for the GWAS marker) (S6 Table). Only 3% of ticked dogs had one copy of the duplication-associated haplotypes (11 heterozygotes and no duplication homozygote). Although the association between ticking and the duplication was significant (chi-squared test, $P = 5.9 \times 10^{-5}$), this could be due to cryptic roaned fur that was not visible in the photographs.

To confirm that the roaned dogs carried the CFA38 tandem duplication, we compared the difference in LRR between a marker located within the duplication (the position 11,140,991) and 10 markers in the flanking region of the duplication ($\Delta LRR = LRR_{inside} - LRR_{outside}$) for dogs with or without roaning pattern. Despite the limited power of $\Delta LRR$ by having only one marker within the duplication, dogs with one or two copies of the duplication-associated haplotypes (i.e., heterozygotes or homozygotes) showed distinct distributions of $\Delta LRR$, supporting the linkage between the duplication and its flanking markers (Fig 5).

The imputed genotypes at CFA38:11,143,243 were nearly perfectly concordant with the duplication genotypes predicted by the haplotypes (S5 Table). All control dogs did not have the duplication (-/-) and were T/T at CFA38:11,143,243 (N = 579). In roaned dogs, 246 dogs carried two copies of the duplication-associated haplotypes and were T/T at CFA38:11,143,243, and 94 dogs carried one copy of the duplication-associated haplotypes and were T/C at CFA38:11,143,243. There were five dogs where the copy number of the duplication-associated haplotypes and the imputed genotype at CFA38:11,143,243 were discordant by having one copy of the duplication-associated haplotypes but were T/T at CFA38:11,143,243 (dog_1448, dog_1685, dog_1691, dog_1981, and dog_2031) (S5 and S6 Tables). They all had one copy of haplotypes that have not been tested by the breakpoint PCR assay (hap_r04, hap_r36, and hap_r38). These untested haplotypes shared a part of haplotypes with hap_r05, hap_GG01, and hap_GG02, respectively, indicating the presence of the duplication in these recombinant haplotypes (S7 Fig and S5 Table).

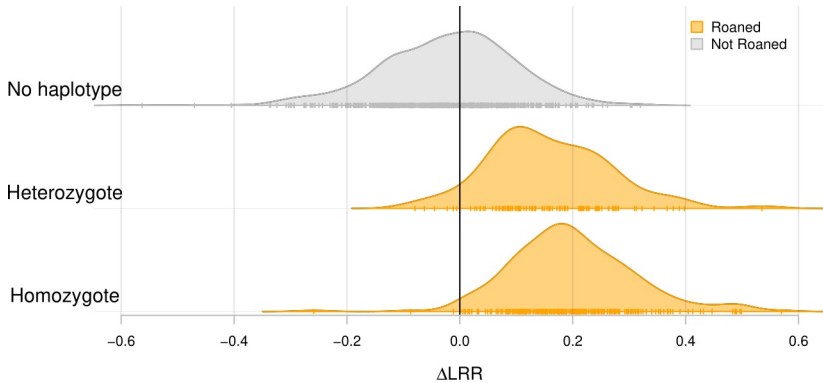

**Fig 5. Density distribution of the array signal intensity (ΔLRR) for the discovery panel dogs with zero, one, or two copies of the duplication-associated haplotypes (no haplotype, heterozygote, and homozygote, respectively).** Vertical ticks indicate individual ΔLRR of dogs with roaning (orange) and without roaning (grey).

## Prediction of roaning coat pattern in the independent validation panel

We selected an independent dataset of 274 mixed breed dogs (120 roaned and 154 control dogs, respectively) (S5 Table) to validate the association between roan coat and the markers on CFA38 (the duplication-associated haplotypes and imputed genotypes of CFA38:11,143,243). All dogs with roaning had at least one copy of the duplication-associated haplotypes and the roan-associated T allele at CFA38:11,143,243, agreeing with the pattern identified in the discovery dataset (S7 Table). In the control group, the duplication was absent in 97% of dogs, whereas we found five dogs without roaned fur that were heterozygous for the duplication. ΔLRR of these five dogs were all positive (0.08–0.50) (S10 Fig), supporting the existence of the duplication in these dogs' genomes. Consistent with the duplication, all of these five dogs were C/T heterozygotes at CFA38:11,143,243 based on genotype imputation (S7 Table). We confirmed the imputed genotypes of four dogs by Sanger sequencing the region CFA38:11,143,161–11,143,326). They had either small spots (or ticking), faint roaning pattern in muzzle areas, a limited amount of white marking (i.e., a possible "residual white"), wolf-like sable pattern without large patches of roaning, or long fur that resulted in inaccurate phenotyping (S11 Fig).

## Selection on the CFA38

Whole-genome re-sequencing data showed that nucleotide diversity ($\pi$) was reduced near the duplication on CFA38 in a breed almost fixed for roaning (Wirehaired Pointing Griffon), whereas such reduction was not observed in two reference breeds, Border Collies and Labrador Retrievers (Fig 6A). Genetic differentiation, measured as $F_{ST}$, was elevated in this region in the comparison between Wirehaired Pointing Griffon and Border Collies but not between Labrador Retrievers and Border Collies, suggesting selection most likely acting on the duplication (Fig 6B). Consistent with the selection signals detected in the whole-genome data, the genotype data of the SNP array revealed that about 50% of Australian Cattle Dogs with roaned coat were autozygous between 10 and 11 Mb on CFA38 (Fig 6C). Similarly, frequent autozygosity was found in Dalmatians but not in Border Collies in this region, suggesting that the duplication-associated haplotype was likely favored by selection in Australian Cattle Dogs and Dalmatians. Moreover, cross-population extended haplotype homozygosity (XP-EHH) [33,34] showed a significant difference between Australian Cattle Dogs and Border Collies, where the maximum XP-EHH of 2.05 was found at a marker position 10,871,209 (Fig 6D). A comparison between Dalmatians and Border Collies also showed a high XP-EHH score in this region (2.56 at the position 11,164,866), consistent with the elevated autozygosity in Dalmatians.

 To infer the frequency of the CFA38 duplication in dog and other canine populations, we searched for the duplication-associated haplotypes, found in the discovery dataset (Figs 3 and S7 and S5 Table), in the WGS dataset with 722 dogs and other canid species [31]. In addition to the breeds that we have used for the discovery of the duplication (Fig 3), 16 breeds had at least one copy of the duplication-associated haplotypes (S8 Table). These haplotypes were fairly common in some breeds, such as German Shepherds Dogs (5 out of 15 dogs) and Belgian Tervurens (4 out of 11 dogs); however, roaning, if any, should not be visible in these breeds because of the lack of white areas (i.e., S/S genotype at S-locus). The duplication-associated haplotypes were also found in breeds where roaning was occasionally observed: Portuguese Water Dogs (3 out of 11 dogs), Lagotto Romagnolos (2 out of 5 dogs), and Dachshunds (2 out of 5 dogs). Finally, village dogs in China, Papua New Guinea, and Vietnam also had the duplication-associated haplotypes (6 out of 45 dogs), indicating a potentially ancient origin of the duplication. Mapped sequence read coverage within the duplication (CFA38:11,131,835–11,143,237) was about 1.5 times and 2 times higher than the surrounding 100-kb flanking

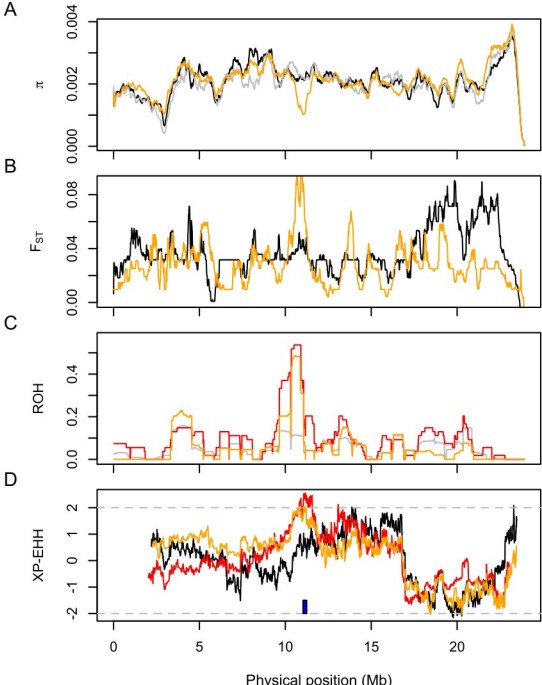

**Fig 6. Signature of selection in the region on CFA38 associated with roaning.** A) Nucleotide diversity ($\pi$) for Wirehaired Pointing Griffon (orange), Border Collies (grey), and Labrador Retriever (black) in 500-kb sliding windows. B) Pairwise genetic difference ($F_{ST}$) for Wirehaired Pointing Griffon (orange) and Labrador Retriever (black). Border Collies were used as a reference. C) ROH in Australian Cattle Dog (orange), Dalmatians (red), and Border Collies (grey). D) XP-EHH in Australian Cattle Dog (orange), Dalmatians (red), and Labrador Retrievers (black). Border Collies were used as a reference. Wirehaired Pointing Griffons and Australian Cattle Dogs are breeds where roaning is common. Blue rectangle: position of the 11-kb duplication. $\pi$ and $F_{ST}$ are estimated by using whole-genome re-sequencing data, while ROH and XP-EHH were estimated by using Embark genotyping data.

region in dogs with one or two copies of the duplication-associated haplotypes, respectively, confirming the association between the haplotypes and the duplication in these breeds (S12 Fig). The duplication-associated haplotypes were not found in Grey Wolves (N = 45).

## Dalmatian spot modifier

F-locus associated with Dalmatian's spots has been shown to be genetically linked to *SLC2A9* on CFA3, a gene responsible for a high level of uric acid excretion in the urine (hyperurico-suria) [8,19]. The segregation pattern of the Dalmatian-like spots in a backcross population of Dalmatians and English Pointers suggests that this phenotype is recessive: *F/-* with flecking (scattered white furs within pigmented region) and *f/f* without flecking [8]. Given the presence of the duplication at the R-locus on CFA38 in both Dalmatians and roaned dogs, we assumed that the recessive *f* allele was fixed in Dalmatians but was rare in roaned dogs. To identify genomic regions associated with the Dalmatian's spot, we imputed genotypes of 604,843 SNVs on CFA3 reported in Plassais *et al.* [31] for 358 roaned dogs in the discovery dataset and 262 Dalmatians (S1 Text). A SNV at CFA3:72,316,930, which was 2.9 Mb away from the causal variant of hyperuricosuria (CFA:69,456,869), showed the largest genotype frequency difference between Dalmatians and roaned dogs, where 98% of dalmatians and 2% of roaned dogs were AA homozygotes (chi square test, p = 1 x 10$^{-120}$) (S9 Table). Six Dalmatians were GA heterozy-gotes, and none of them were homozygous for the reference G allele. This marker was in an intronic region of the Ras Homolog Family Member H (*RHOH*) gene.

To test if the candidate marker CFA3:72,316,930 was associated with Dalmatian-like spots, we additionally genotyped 43 mixed breed dogs with 1) Dalmatian-like spots without flecking or other types of spots with flecking, 2) white background coat with possibly $s^w/s^w$ extreme white genotype at S-locus, and 3) at least one copy of the duplication-associated haplotype on CFA38 (S10 Table). Seven dogs had typical Dalmatian-like spots in most of their bodies, while the remaining 36 dogs had black or brown ticks smaller than typical Dalmatian spots usually with some white hairs interspersed within the ticks (S13 Fig). Similar to the purebred Dalmatians, imputed genotypes at CFA3:72,316,930 of seven mixed breed dogs with Dalmatian-like spots were all AA homozygous. However, four dogs without Dalmatian-like spots also had AA homozygous genotype (S5 Table). We did not find SVs in this region in the two Dalmatians in the WGS dataset.

## Discussion

### A novel locus associated with roaning but not strongly with ticking

It has been debated whether T-locus for ticking and R-locus for roaning have a similar molecular basis since the mid 20th century [40]. Although the results of this study did not yield genomic regions associated with ticking, we showed that a duplication at 11.13–11.14 Mb on CFA38 was strongly associated with roaning, implying that these two theoretical loci are likely located in different genomic regions. Owing to the extended linkage in the surrounding region of the duplication, we were able to identify haplotypes with the duplication. The presence of the duplication in these haplotypes was confirmed by the breakpoint PCR assay, Sanger sequencing of the PCR amplicon spanning the duplication midpoint, and whole-genome resequencing data for the identification of discordant read pairs and abrupt read depth increase. In addition, the distribution of the array signal intensity in dogs with 0, 1, or 2 copies of the duplication-associated haplotypes was in agreement with the expected distribution. This mutation is nearly completely penetrant by explaining more than 99% of roaning cases in both purebred and mixed breed dogs. Thus, our haplotype-based linkage test can accurately detect the presence of the CFA38 duplication, which has high predictability for the roaning coat pattern. The strong association between the phenotype and the imputed SNP genotype at CFA38:11,143,243, which is tightly linked with the duplication, further supports the involvement of this region in this coat pattern.

Roaning was found in dogs with one or two copies of the duplication-associated haplotypes, suggesting that the CFA38 duplication is dominant in its effect on fur pigmentation. Pigmented fur in a roaned coat is intermingled with unpigmented fur, which is associated with certain S-locus variants, including the SINE insertion and variable length polymorphisms in the putative promoter region of *MITF* on CFA20 [17,18]. Although our SNP genotyping array platform could not unambiguously distinguish all of these S-locus variants, the presence of all three genotypes at the SINE indel marker (CFA20:21,836,232) in roaned dogs (S5 Table) suggests that the association between roaning and the duplication on CFA38 is robust regardless of the type of white spotting patterns (i.e., Irish spotting, piebald, and extreme white). We found a total of five non-roaned dogs with one copy of the duplication-associated haplotypes in the validation panels. Four of them (dog_10056, dog_10079, dog_10087, and dog_10166) had a long coat, which makes it difficult to accurately distinguish between ticked and roaned patterning, while the remaining dog had limited white spotting patterns (dog_10028). The small white spotting pattern is likely a residual white, which we excluded from the study (see Methods). Assuming that the phenotypes of these dogs were correctly assigned, there might be additional modifier loci interacting with R-locus and/or S-locus.

Our VEP analysis did not find any SNVs and indels that could potentially be functionally relevant to pigmentation in the roan-associated 119-kb region. This left the 11-kb tandem duplication as the strongest candidate for roaning, but follow-up functional validation is needed to investigate the exact impact of this duplication on the development of pigmented fur in otherwise unpigmented area. Our comparative analysis identified putative regulatory regions within and nearby the duplication that are highly conserved in vertebrates (S9 Fig). *USH2A* encodes the protein *usherin*. In humans, *USH2A* mutations are associated with Usher syndrome, characterized by progressive hearing loss and vision impairment often accompanied by retinitis pigmentosa [41]. A functional assay by using *USH2A* knockout mice showed that this gene is involved in the maintenance of retinal photoreceptors and the development of cochlear (inner ear) hair cells [42]. A recent study showed that a mutation in *USH2A* showed abnormal pigment deposition and reduced expression of *MITF* and other melanin metabolism-related genes, such as tyrosinase (*TYR*) and oculocutaneous albinism II (*OCA2*) genes, in retinal cells derived from induced pluripotent stem cells, indicating a potential involvement of *USH2A* in the pigmentation pathway [43]. Interestingly, the distribution of usherin in healthy individuals is highly conserved between mice and humans, in which skin was completely devoid of this protein [44,45]. We speculate that the duplication of the putative regulatory regions may result in ectopic expression of *USH2A* in skin melanocytes. Alternatively, the duplication may facilitate alternative splicing and create a novel protein isoform since this complex gene with 73 exons is known to form several isoforms [41,46]. High prevalence of congenital deafness in Dalmatians, Australian Cattle Dogs, and other commonly roaned breeds [47] may also imply a potential pleiotropic effect of *USH2A* on both hearing and pigmentation [48].

### Evolutionary origin of roaning

The strong selection signals in the surrounding region of the CFA38 duplication suggest that the frequency of the duplication increased during the formation of modern breeds (Fig 6). In our discovery panel dogs, the duplication is common in Australian Cattle Dog (94%), Cesky Fousek (100%), English Cocker Spaniel (100%), German Shorthaired Pointer (73%), German Wirehaired Pointer (100%), Spinone Italiano (100%), and Wirehaired Pointing Griffon (100%) (S1 Table). In addition, the duplication-associated haplotypes were found in other distantly-related breeds (e.g., German Shepherd Dogs and Portuguese Water Dogs) and village dogs (i.e., indigenous dogs that accompany humans but are not selectively bred), indicating that selection acted on a variation that existed in the ancestral canine population (i.e., "soft sweep").

Because of the high prevalence of the CFA38 duplication in Dalmatians, we hypothesize that Dalmatian's spots have a similar genetic basis with roaning, while the interaction between the CFA38 duplication and F-locus on CFA3 localizes the distribution of pigmented fur. This is in line with the development pattern of spots in Dalmatians and roaning, both of which emerge between two to eight weeks after birth. While we cannot completely rule out an involvement of uncharacterized T-locus, the shared haplotype on CFA38 (Fig 3) and a nested phylogenetic position of Dalmatians within commonly roaned breeds (e.g., English Cocker Spaniel and German Wirehaired Pointer) [49] suggests that Dalmatian's spots have been likely derived from roaned ancestors.

The Australian Cattle Dog, as its name suggests, was established in Australia in the 19th century by crossing some Collie-type dogs with Dingos (a wild dog in Australia), Bull Terriers, Kelpies and Dalmatians [50]. Therefore, we believe that the duplication-associated haplotypes on CFA38 were introgressed from Dalmatians to the ancestral population of Australian Cattle Dog during its breed formation. Following selective breeding for roaning likely increased the frequency of the duplication in the Australia Cattle Dog population.

In conclusion, we showed a potential involvement of *USH2A* in coat pigmentation in dogs by using genome-wide SNP array genotyping and customer-provided photographs. An 11-kb duplication in the intronic region of this gene was strongly associated with roaned coat in both purebred and mixed breed dogs. Since there was no association signal near *KITLG* on CFA15, we confirmed an earlier notion where *KITLG* was not responsible for roaning in dogs [9]. This study provides another example of phenotypic convergence where roaned coat has evolved independently in dogs, horses, cattles, among others by using different genes. The similar genetic makeup of Dalmatian's spots and roaning in other breeds affirms the important role of epistatic interaction in the evolution of novel phenotype. In addition, a tentative causal mutation lies in a non-coding region which may modify expression patterns of *USH2A*. Our study highlights the importance of epistatic interactions and rewiring regulatory networks as a contributor to a burst of phenotypic divergence.

## Supporting information

**S1 Fig. Representative dogs with the different extent of ticking in herding breeds (top row) and non-herding breeds (bottom row).** A) and G) No ticking. B), C), H), and I) Lightly ticked. D) and J) Heavily ticked. E), F), K) and L) Heavily ticked but also roaned. Dogs with ticking and roaning were excluded from the study. Photo credit (from A to L): Gusto and Chris Cameron Sarafin, Hootenanny and Deanna Haase, Lira and Judi Schachte, Hugo and Kirby Brannon, Jetta and W.Fluckey/C.Moore, Darwin and Colin Scott, Paco Muñoz-Scaggs and Diana Muñoz-Scaggs, Martin and Brittney Mitchell, Nova Fox Bijou Duchesse de Hedwige and K. Smith, Joey and Marissa Crean, Hans and Kari Cueva, CH Cedar Creek's I'm So Fancy and Brittany Fischer.
(JPG)

**S2 Fig. Representative coat phenotypes.** A) Spinone Italiano (roaned). B) Australian Cattle Dog (roaned). C) Stabyhoun (ticked). D) Border Collie (ticked). E) German Shorthaired Pointer (both roaned and ticked). F) Australian Cattle Dog (both roaned and ticked). G) Pointer (without roaning and ticking). H) Border Collie (without roaning and ticking). A, C, E, and G are non- herding breeds, while B, D, F, and H are herding breeds. Photo credit (from A to H): A. Barber, Alexis Q., E. Nado, Heather O'Neill, Chris and Barbara T., G. C., Erica Murray, and Adriana N.
(JPG)

**S3 Fig.** Q-Q plots of the association with A) roaning and B) ticking.
(PNG)

**S4 Fig. Manhattan plots of association with roaning.** A) Herding breeds. B) Non-herding breeds. Red and blue horizontal lines are significant ($P < 5 \times 10^{-8}$) and suggestive ($P < 1 \times 10^{-5}$) associations, respectively.
(PNG)

**S5 Fig. Manhattan plots of association with ticking.** A) Herding breeds. B) Non-herding breeds. Red and blue horizontal lines are significant ($P < 5 \times 10^{-8}$) and suggestive ($P < 1 \times 10^{-5}$) associations, respectively.
(PNG)

**S6 Fig.** Discordant read pairs at the duplication breakpoint on CFA38 identified in Australian Cattle Dog (top panel), and Border Collie (bottom panel). Outward-facing read pairs (green) indicate that this is a tandem duplication found in Australian Cattle Dog (usually roaned) but not in Border Collie.
(PDF)

**S7 Fig. Haplotypes in the roan-associated region on chromosome 38.** A) Manhattan plot of association with roaning on chromosome 38. Red and blue horizontal lines are significant ($P < 5$ x $10^{-8}$) and suggestive ($P < 1$ x $10^{-5}$) associations, respectively. Genes are indicated in grey boxes. Grey vertical lines indicate the region indicated in the panel B. B) Haplotypes defined by 52 single nucleotide variant (SNV) markers in the roan-associated region (CFA38:10,985,456–11,380,922) and their frequencies. Rows correspond to haplotypes, and columns correspond to markers. Haplotypes in orange (hap_01—hap_21) have the roan-associated "A" allele at the most significant marker (CFA38:11,085,443). Eight haplotypes in yellow are identified in roaned dogs without the roan-associated "A" allele at the most significant marker. A haplotype in red was found in Dalmatians. Haplotypes in grey were identified in control groups without roaning. Only the haplotypes that are mentioned in the main text or with frequencies larger than 10 are shown (see S5 Table for the full list of haplotypes).
(PDF)

**S8 Fig. Manhattan plot of association with roaning and ticking based on imputed genotypes on CFA38.** A) Entire chromosome. B) A region close to the most significant marker (magenta circle). The most significant GWAS marker is in red circle. The most significant CWAS marker is in magenta circle. Red and blue horizontal lines are significant ($P < 5$ x $10^{-8}$) and suggestive ($P < 1$ x $10^{-5}$) associations, respectively.
(PDF)

**S9 Fig. Human orthologous region (hg38) of the CFA38 associated with roaning (UCSC genome browser).** The highlighted area in blue is the orthologous region to the tandem duplication identified in dogs with roaning, which is located within the intron 61 of *USH2A*. Gene-Hancer Regulatory Elements are located at chr1:215,715,579–215,717,032 (green line), which corresponds to CFA38:11,146,170–11,147,605 in the dog genome (CanFam3.1). DNAse I hypersensitive sites: grey and black boxes. Open Regulatory Annotation (ORegAnno): orange and blue boxes. Cons 100 Verts (100 vertebrates basewise conservation by PhyloP score): blue histogram.
(PNG)

**S10 Fig. Density distribution of the array signal intensity (ΔLRR) for the validation panel dogs with zero, one, or two copies of the duplication-associated haplotypes (no haplotype, heterozygote, and homozygote, respectively).** Vertical ticks indicate individual ΔLRR of dogs with roaning (orange) and without roaning (grey). Density plots with the number of individuals less than 10 are not shown, but individual ΔLRR is indicated with longer vertical ticks.
(PDF)

**S11 Fig. Coat phenotypes of mixed breed dogs in the validation dataset.** These five dogs carry a duplication-associated haplotype but show no or little roaned coat. In the panel E, roan and/or tick is invisible when fur is long (left) but is visible when the coat is shaved (right). Photo credit (from A to E): William DeLozier, Dorothy Olszewski, Thomas Borr, Rebeccah Kivitz, and Maria Casey.
(PNG)

**S12 Fig. Violin plot showing the density distribution of the normalized read depth in the duplicated region on CFA38.** Mean variant read depth (DP) within the duplication was divided by the mean variant DP of the flanking 100-kb region for normalization.
(PDF)

**S13 Fig. Representative coat phenotypes of mixed breed dogs with and without Dalmatian-like spots.** Photo credit (clockwise from top-left): Gina Vrdoljak, Lindsay Jakobovits, Darragh Nolan, Catharine Giannasi, Ann Holland, Susie Johnston, Laurence Montgomery, and Raquel Lorenzo.
(JPG)

**S1 Table. The number of dogs and breeds used for genome-wide association study.**
(XLSX)

**S2 Table. Whole-genome re-sequencing data used for characterizing the 11-kb tandem duplication on CFA38.** Data from Plassais et al. (2019) [31].
(XLSX)

**S3 Table. Validation of the CFA38 duplication by PCR and Sanger sequencing.** (A) Primer sequences used for PCR assays described in Fig 4. The expected sizes of the PCR products are 539 bp (Tick38-F1/Tick38-R1), 476 bp (Tick38-F2-2/Tick38-R2-2), and 397 bp (Tick38-F2-2/Tick38-R1). The former two pairs amplify franking region of the CFA38 duplication (positive controls), whereas the last pair amplifies the midpoint of the duplication. (B) Midpoint span product sequence. Nucleotides in bold and italic are likely the end of the first copy and the beginning of the second copy, respectively.
(DOCX)

**S4 Table. Primer sequences used for PCR to genotype SNPs associated with coat color phenotype.** (A) The top CWAS marker associated with roaning (CFA38:11,143,243). (B) A candidate F-locus marker associated with Dalmatian's spot (CFA3:72,316,930).
(DOCX)

**S5 Table. Haplotypes identified in roaned dogs and control dogs in the discovery and validation panels as well as purebred Dalmatians.** The following results are also included: genotypes at the most significant marker in GWAS (CFA38:11,085,443) (GWAS_marker), the copy number of the duplicated region (CFA38:11,131,835–11,143–237) inferred by haplotypes (Dup_CN), presence/absence (1 or 0) of the 400-bp amplicon in the breakpoint PCR assay (BP_PCR), imputed genotypes at the most significant marker in CWAS (CFA38:11,143,243) (CWAS_marker) and their validation result by Sanger sequencing (CWAS_sanger), and genotypes of the marker for the presence/absence of the SINE element in a putative regulatory region of MITF (CFA20:21,836,232).
(XLSX)

**S6 Table. Genotype frequencies of the markers associated with roaning in the discovery panel.** A) CFA38 Duplication. B) The top associated CWAS marker based on the imputed genotypes (CFA38:11,143,243). C) The top associated GWAS marker (CFA38:11,085,443). D) CFA38 Duplication and the top associated CWAS in roaned dogs. E) A missense mutation at CFA38:11,111,286 based on the imputed genotypes. F) A missense mutation at CFA38:11,169,445 based on the imputed genotypes.
(DOCX)

**S7 Table. Genotype frequencies of the CFA38 duplication associated with roaning and the imputed genotype at CFA38:11,143,243 in the validation panel.**
(DOCX)

**S8 Table. Frequency of the duplication-associated haplotypes identified in 722 dogs and canid species with the whole-genome resequencing data.**
(XLSX)

**S9 Table. Genotype frequencies at the causal variant of hyperuricosuria at CFA3:69,456,869 and CFA3:72,316,930 in purebred Dalmatian, roaned dogs, and mixed breeds with or without Dalmatian-like spots.** A) Genotype frequency at CFA3:69,456,869. B) Genotype frequency at CFA3:72,316,930 based on the imputed genotypes. Roaned dogs were purepred dogs used for GWAS (the discovery panel dogs).
(DOCX)

**S10 Table. Imputed genotypes at the most significant marker in CWAS (CFA38:11,143,243) (CFA3_marker) in roaned dogs in the discovery panel, purebred Dalmatians, and mixed breed dogs with or without Dalmatian-like spots.** The following results are also included: genotypes at the most significant marker in roaning GWAS (CFA38:11,085,443) (GWAS_marker), imputed genotypes at the most significant marker in roaning CWAS (CFA38:11,143,243) (CWAS_marker), and the copy number of the duplicated region (CFA38:11,131,835–11,143–237) inferred by haplotypes (Dup_CN).
(XLSX)

**S1 File. A list of 220,484 markers used in the study.** Columns are chromosome, marker ID, genetic position (cM), physical position (bp), allele 1, and allele 2.
(ZIP)

**S2 File. Genome wide association between markers and roaning in the discovery panel.**
(ZIP)

**S3 File. Genome wide association between markers and roaning in the discovery panel (herding breeds).**
(ZIP)

**S4 File. Genome wide association between markers and roaning in the discovery panel (non-herding breeds).**
(ZIP)

**S5 File. Genome wide association between markers and ticking in the discovery panel.**
(ZIP)

**S6 File. Genome wide association between markers and ticking in the discovery panel (herding breeds).**
(ZIP)

**S7 File. Genome wide association between markers and ticking in the discovery panel (non-herding breeds).**
(ZIP)

**S8 File. Chromosome wide association between markers with imputed genotypes on chromosome 38 and roaning in the discovery panel.**
(ZIP)

**S9 File. Potential functional impact of 67 variants in the roan-associated region on CFA38 predicted by Variant Effect Predictor (VEP).** Allele frequencies (A1_freq and A2_freq) are based on 16 potentially roaned dogs in the WGS dataset.
(ZIP)

**S1 Text. Genotype-phenotype association analysis by using imputed genotypes.**
(PDF)

## Acknowledgments

We thank all Embark customers who agreed to participate in our research. We thank the following dog owners who allowed us to use their dogs' photographs in this manuscript: A. Barber and Fig; Kirby Brannon and Hugo; L. Bray and Ralph; G. C. and Hotch; Marissa Crean and Joey; Kari Cueva and Hans; Dominic Ebacher and Index; Brittany Fischer and CH Cedar Creek's I'm So Fancy; W. Fluckey and C. Moore and Jetta; Kellina H. and Miss Zoey; Deanna Haase and Hootenanny; Lisa Hayden and "Scout" Ch Cathedral's Head Over Heels; Fernanda Lesnau and Kisfarkas Explosion Black Diesel; Brittney Mitchell and Martin; Diana Muñoz-Scaggs and Paco Muñoz-Scaggs; Erica Murray and Sam; Adriana N. and Gambit; Hanna N. and Calvary's Mountain Sound; E. Nado and Finnley Kisses Wade call name "Breeze"; Heather O'Neill and Chip; Alexis Q. and Koda; Alison Ruhe and Jazz; and Chris and Barbara T. and Lillie; Chris Cameron Sarafin and Gusto; Judi Schachte and Lira; Colin Scott and Darwin; K. Smith and Nova Fox Bijou Duchesse de Hedwige. We thank all the Embark employees who made this work possible, particularly Matt Barton, Stephen Fromm, Adam Gardner, and Alan Yang for developing bioinformatics infrastructure, and Casey Ehrlich, Alison Ruhe, and Ashley Troutman for their suggestions and feedback on customer communication and breed-specific traits.

## Author Contributions

**Conceptualization:** Takeshi Kawakami, Meghan K. Jensen, Andrea Slavney, Petra E. Deane, Erin T. Chu, Aaron J. Sams, Adam R. Boyko.

**Data curation:** Takeshi Kawakami, Meghan K. Jensen.

**Formal analysis:** Takeshi Kawakami, Meghan K. Jensen, Andrea Slavney, Ausra Milano, Vandana Raghavan, Brett Ford.

**Investigation:** Meghan K. Jensen, Andrea Slavney, Petra E. Deane, Brett Ford, Erin T. Chu.

**Methodology:** Takeshi Kawakami, Meghan K. Jensen, Andrea Slavney, Petra E. Deane, Ausra Milano, Erin T. Chu.

**Project administration:** Aaron J. Sams, Adam R. Boyko.

**Supervision:** Erin T. Chu, Aaron J. Sams, Adam R. Boyko.

**Validation:** Vandana Raghavan.

**Visualization:** Takeshi Kawakami, Meghan K. Jensen, Vandana Raghavan.

**Writing – original draft:** Takeshi Kawakami.

**Writing – review & editing:** Takeshi Kawakami, Meghan K. Jensen, Andrea Slavney, Petra E. Deane, Brett Ford, Erin T. Chu, Aaron J. Sams, Adam R. Boyko.

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
