## [Decision Letter · Decision Letter 0]

27 Jan 2021

PONE-D-20-35246

R-locus for roaned coat is associated with a tandem duplication in an intronic region of USH2A in dogs and also contributes to Dalmatian spotting

PLOS ONE

Dear Dr. Kawakami,

Thank you for submitting your manuscript to PLOS ONE. After careful consideration, we feel that it has merit but does not fully meet PLOS ONE’s publication criteria as it currently stands. Therefore, we invite you to submit a revised version of the manuscript that addresses the points raised during the review process.

(As an AE, I would like to apologize for the delay in evaluating your submission - it was very difficult to find reviewers over the holiday period. )

Reviewer 1 raises several valid criticisms and pertinent comments that need to be addressed during the revision of your manuscript. Reviewer 1 would also like to see more follow-up experiments, incl. functional validation, of the variant. While I agree that it would be of great value to see such a validation and more information on the biological underpinnings of this variant and its phenotypic effects, I will not consider such experiments as a requirement for a successful manuscript revision. Your main result that the USH2A variant is strongly associated with the roaning phenotype is in itself a stand-alone and robust result (cf. PLOS ONE’s publication criteria). In addition, I suggest you follow the advice of reviewer 1 by (a) reworking (reducing) the discussion of the effects on flecking, as not much can be said at this stage and (b) evaluate more specifically biological hypotheses on how the USH2A variant may exert its effects. Importantly, please check that all data used for/generated in this study can be accessed.

We look forward to receiving your revised manuscript.

Kind regards,

Christian Braendle

Academic Editor

PLOS ONE

Journal Requirements:

2. We note that you are reporting an analysis of a microarray, next-generation sequencing, or deep sequencing data set. PLOS requires that authors comply with field-specific standards for preparation, recording, and deposition of data in repositories appropriate to their field. Please upload these data to a stable, public repository (such as ArrayExpress, Gene Expression Omnibus (GEO), DNA Data Bank of Japan (DDBJ), NCBI GenBank, NCBI Sequence Read Archive, or EMBL Nucleotide Sequence Database (ENA)). In your revised cover letter, please provide the relevant accession numbers that may be used to access these data. For a full list of recommended repositories, see http://journals.plos.org/plosone/s/data-availability#loc-omics or http://journals.plos.org/plosone/s/data-availability#loc-sequencing

3. Thank you for providing the following Funding Statement: 

'This study was funded by Embark Veterinary, Inc. and the participants that provided DNA and phenotypic information via Embark’s web-based platform.'

a. We note that one or more of the authors is affiliated with the funding organization, indicating the funder may have had some role in the design, data collection, analysis or preparation of your manuscript for publication; in other words, the funder played an indirect role through the participation of the co-authors.

If the funding organization did not play a role in the study design, data collection and analysis, decision to publish, or preparation of the manuscript and only provided financial support in the form of authors' salaries and/or research materials, please review your statements relating to the author contributions, and ensure you have specifically and accurately indicated the role(s) that these authors had in your study in the Author Contributions section of the online submission form. Please make any necessary amendments directly within this section of the online submission form. 

Please also update your Funding Statement to include the following statement: “The funder provided support in the form of salaries for authors [insert relevant initials], but did not have any additional role in the study design, data collection and analysis, decision to publish, or preparation of the manuscript. The specific roles of these authors are articulated in the ‘author contributions’ section.”

If the funding organization did have an additional role, please state and explain that role within your Funding Statement.

b. We note that one or more of the authors are employed by commercial companies: Mascoma LLC Lallemand Corporation and Amazon Web Services, Inc.

Please ensure that you declare these commercial affiliations in the amended Funding Statement , as well as a statement regarding the Role of Funders in your study.

c. Please also provide an updated Competing Interests Statement declaring these commercial affiliations along with any other relevant declarations relating to employment, consultancy, patents, products in development, or marketed products, etc.  

Within your Competing Interests Statement, please confirm that these commercial affiliations do not alter your adherence to all PLOS ONE policies on sharing data and materials by including the following statement: "This does not alter our adherence to  PLOS ONE policies on sharing data and materials.” (as detailed online in our guide for authors http://journals.plos.org/plosone/s/competing-interests). If this adherence statement is not accurate and  there are restrictions on sharing of data and/or materials, please state these. Please note that we cannot proceed with consideration of your article until this information has been declared.

Reviewers' comments:

Reviewer's Responses to Questions

**Comments to the Author**

1. Is the manuscript technically sound, and do the data support the conclusions?

Reviewer #1: Partly

Reviewer #2: Yes

2. Has the statistical analysis been performed appropriately and rigorously? 

Reviewer #1: Yes

Reviewer #2: Yes

3. Have the authors made all data underlying the findings in their manuscript fully available?

Reviewer #1: No

Reviewer #2: Yes

4. Is the manuscript presented in an intelligible fashion and written in standard English?

Reviewer #1: Yes

Reviewer #2: Yes

5. Review Comments to the Author

Reviewer #1: Kawakami et al. utilized dog samples and photographs that were submitted to Embark for commercial genetic profiling. The authors used the data to investigate the genetic basis of several canine coat color phenotypes, roaning, ticking, and flecking. The authors performed GWAS and identified an association signal for roaning on chromosome 38. Further analyses revealed an intronic 11 kb duplication in the USH2A gene encoding usherin, which is very strongly associated with the roaning phenotype. Somewhat surprisingly, the authors did not detect any association signal for ticking. The analysis on flecking is very indirect and circumstantial. It is based on the hypothesis that flecking in Dalmatians is linked to the hyperuricosuria locus, which was published in 1976 (Schaible et al.), before the availability of genetic markers. I was not able to access the full text of this 1976 publication. Given that the authors also show that the previous assumption on an interaction between ticking and flecking in Dalmatian spotting is apparently not true, I consider the work on flecking as too superficial and premature to be published at this time. The main result of the study is the identification of a plausible candidate cuasative variant for roaning. Demonstrating a causal role of USH2A for this phenotype would be an important finding of broad scientific interest. Unfortunately, the authors only report in (too) great detail about the identification of the variant, but provide no functional follow-up or at least a plausible mechanistic hypothesis how the duplication in USH2A might exert its effect.

Major comments:

(1) Some functional confirmation experiments are essential. Is there a qualitative (splicing) or quantitative effect on USH2A splicing? This should not be too difficult to assess in heterozygous animals where the transcripts from the wildtype and mutant allele can be directly compared to each other (e.g. by RNA-seq).

(2)

At least a hypothesis should be presented how a dominant USH2A allele (gain of function?) might lead to increased pigmentation in the "white" patches of white spotted dogs.

(3)

The manuscript gives a huge amount of details on various imputation procedures, haplotype phasing, and other means to extract maximum information from the existing SNP chip genotypes. While this is admittedly very important for a diagnostic lab, it yields little insight into the biology of the trait. I wonder whether the manuscript could be restructured in a way that the manuscript also becomes informative and interesting to readers with a more focussed interest in pigmentation biology. Could some of the technical/diagnostic details be moved to supplementary data (perhaps as a supplemenatry methods text)?

(4)

Lines 380-382: "Since Variant Effect Predictor (VEP) [25] suggested that none of these variants had a large impact on the function of USH2A...". This statement is not true. VEP predicted several missense and frameshift variants. These were only excluded due to their genotype distribution, not due to the VEP prediction. The presentation of the manuscript makes it very difficult to follow the argumentation of the authors. The chapter "functional annotation" should be placed here (~line 380), before the search for structural variants.

(5)

The work on flecking is too premature and inconclusive. The authors either need to present more complete data including a plausible mechanistic hypothesis how the flecking allele could work together with the roaning allele to produce the large solid spots in Dalmations or the entire sections should be reduced to a very short statement that the data on flecking are inconclusive (similar to the situation with ticking).

(6)

File S1: PLoS journals require deposition of the complete raw data. The file with the marker names (bim-file) is insufficient. The authors should make bed-, bim-, and fam-files available or alternatively a ped and map file.

Minor comments:

(7)

Line 81: Roan in horses is linked to the KIT locus, not the KITLG locus.

(8)

Line 207: Insert an "and" between roaned and 567.

(9)

Line 541: 120 roaned and 154 control dogs

(10)

Line 618: It does not make sense to give amino acid exchanges for chromosomal (genomic) positions. The authors should give the amino acid variants with respect to specific proteins. HGVS nomenclature rules should be followed (https://varnomen.hgvs.org/).

(11)

Figure S8: It should be explicitly stated what the marker in the magenta circle is. Is this the 11 kb duplication?

(12)

Table S7: This table either needs much more detailed explanation or it can be deleted entirely. How do the authors determine K^B vs K^y when it is unknwon whether any K^br alleles are present? This will be difficult with an illumina array. To the best of my knowledge there are at least 3 different published e-alleles and 5 different b-alleles. The causal variant underlying the various agouti alleles ahve not been published (or under dispute). If this table shall remain in the manuscript, it must be explicitly stated which markers were genotyped to predict the coat color genotypes.

(13)

File S9: This is a relatively small table. This should be given as a Supplementary table (Excel-file), rather than as a compressed zip-file.

Reviewer #2: The authors use citizen science provided by dog owners in the form of photos of their dogs to find molecular explanation for coat color patterning in dogs. This is a lengthy study and authors mostly do a great job in describing this study in detail. The manuscript is technically sound, and data support the conclusions. The authors have made data underlining the data adequately available. The manuscript is written in standard English.

Minor comments:

1) The authors have had various breeds in the non-roan control group that should be mentioned and instead not include at all the breed Labrador Retriever that would be phenotypically always non-roan despite of the roaning genotype it might have due to this breed not presenting any color patches of white were the roan could be observed.

2) Chromosome-wide association analysis (CWAS) is incorrectly abbreviated.

6. PLOS authors have the option to publish the peer review history of their article (what does this mean?). If published, this will include your full peer review and any attached files.

Reviewer #1: No

Reviewer #2: No

---

## [Author Response · Author response to Decision Letter 0]

19 Feb 2021

Authors’ responses are indicated in red (AU)

PONE-D-20-35246

R-locus for roaned coat is associated with a tandem duplication in an intronic region of USH2A in dogs and also contributes to Dalmatian spotting

PLOS ONE

Dear Dr. Kawakami,

Thank you for submitting your manuscript to PLOS ONE. After careful consideration, we feel that it has merit but does not fully meet PLOS ONE’s publication criteria as it currently stands. Therefore, we invite you to submit a revised version of the manuscript that addresses the points raised during the review process.

(As an AE, I would like to apologize for the delay in evaluating your submission - it was very difficult to find reviewers over the holiday period. )

Reviewer 1 raises several valid criticisms and pertinent comments that need to be addressed during the revision of your manuscript. Reviewer 1 would also like to see more follow-up experiments, incl. functional validation, of the variant. While I agree that it would be of great value to see such a validation and more information on the biological underpinnings of this variant and its phenotypic effects, I will not consider such experiments as a requirement for a successful manuscript revision. Your main result that the USH2A variant is strongly associated with the roaning phenotype is in itself a stand-alone and robust result (cf. PLOS ONE’s publication criteria). In addition, I suggest you follow the advice of reviewer 1 by (a) reworking (reducing) the discussion of the effects on flecking, as not much can be said at this stage and (b) evaluate more specifically biological hypotheses on how the USH2A variant may exert its effects. Importantly, please check that all data used for/generated in this study can be accessed.

AU: We would like to thank the editor and the reviewers for thoroughly reviewing our manuscript and providing constructive comments. We incorporated reviewer 1’s comment about flecking by reducing the discussion and clarifying the major findings. With regard to the possible effect of USH2A on the process of pigmentation, we revised our discussion by minimizing the amount of speculative discussion. We have provided possible explanations about how USH2A can be involved in the pigmentation pathways in the Discussion (Line 843-865). We have now deposited all the raw genotype data in Dryad, which will be freely available if/after this manuscript is accepted (doi:10.5061/dryad.qz612jmd).

We look forward to receiving your revised manuscript.

Kind regards,

Christian Braendle

Academic Editor

PLOS ONE

Journal Requirements:

AU: We have checked the style requirements and reformatted where necessary.

2. We note that you are reporting an analysis of a microarray, next-generation sequencing, or deep sequencing data set. PLOS requires that authors comply with field-specific standards for preparation, recording, and deposition of data in repositories appropriate to their field. Please upload these data to a stable, public repository (such as ArrayExpress, Gene Expression Omnibus (GEO), DNA Data Bank of Japan (DDBJ), NCBI GenBank, NCBI Sequence Read Archive, or EMBL Nucleotide Sequence Database (ENA)). In your revised cover letter, please provide the relevant accession numbers that may be used to access these data. For a full list of recommended repositories, see http://journals.plos.org/plosone/s/data-availability#loc-omics or http://journals.plos.org/plosone/s/data-availability#loc-sequencing

AU: We have now deposited all the raw data in Dryad, which will be freely available after satisfactory revisions (doi:10.5061/dryad.qz612jmd).

3. Thank you for providing the following Funding Statement: 

'This study was funded by Embark Veterinary, Inc. and the participants that provided DNA and phenotypic information via Embark’s web-based platform.'

a. We note that one or more of the authors is affiliated with the funding organization, indicating the funder may have had some role in the design, data collection, analysis or preparation of your manuscript for publication; in other words, the funder played an indirect role through the participation of the co-authors.

If the funding organization did not play a role in the study design, data collection and analysis, decision to publish, or preparation of the manuscript and only provided financial support in the form of authors' salaries and/or research materials, please review your statements relating to the author contributions, and ensure you have specifically and accurately indicated the role(s) that these authors had in your study in the Author Contributions section of the online submission form. Please make any necessary amendments directly within this section of the online submission form. 

Please also update your Funding Statement to include the following statement: “The funder provided support in the form of salaries for authors [insert relevant initials], but did not have any additional role in the study design, data collection and analysis, decision to publish, or preparation of the manuscript. The specific roles of these authors are articulated in the ‘author contributions’ section.”

If the funding organization did have an additional role, please state and explain that role within your Funding Statement.

AU: We added the following sentence. “The funder only provided financial support in the form of salaries for all authors but did not have any additional role in the study design, data collection and analysis, decision to publish, or preparation of the manuscript.”

b. We note that one or more of the authors are employed by commercial companies: Mascoma LLC Lallemand Corporation and Amazon Web Services, Inc.

Please ensure that you declare these commercial affiliations in the amended Funding Statement , as well as a statement regarding the Role of Funders in your study.

AU: We added the following sentence. “Mascoma LLC Lallemand Corporation and Amazon Web Services only provided financial support in the form of salaries for authors (PED and ETC, respectively) but did not have any additional role in the study design, data collection and analysis, decision to publish, or preparation of the manuscript.”

c. Please also provide an updated Competing Interests Statement declaring these commercial affiliations along with any other relevant declarations relating to employment, consultancy, patents, products in development, or marketed products, etc. 

Within your Competing Interests Statement, please confirm that these commercial affiliations do not alter your adherence to all PLOS ONE policies on sharing data and materials by including the following statement: "This does not alter our adherence to PLOS ONE policies on sharing data and materials.” (as detailed online in our guide for authors http://journals.plos.org/plosone/s/competing-interests). If this adherence statement is not accurate and there are restrictions on sharing of data and/or materials, please state these. Please note that we cannot proceed with consideration of your article until this information has been declared.

 AU: We modified the statement by adding a sentence

“I have read the journal's policy and the authors of this manuscript have the following competing interests: TK, MJ, AS, AM, VR, BF, AJS and ARB are employees of Embark Veterinary, a canine DNA testing company which will offer commercial testing for the variant described in this study. ARB is co-founder and part owner of Embark. PED and ETC were employees of Embark Veterinary when this study was conducted but were employees of Mascoma LLC Lallemand Corporation and Amazon Web Services, respectively by the time of manuscript submission. Mascoma LLC Lallemand Corporation and Amazon Web Services do not have any competing interests with this study and do not alter our adherence to PLOS ONE policies on sharing data and materials.”

Reviewers' comments:

Reviewer's Responses to Questions

Comments to the Author

1. Is the manuscript technically sound, and do the data support the conclusions?

Reviewer #1: Partly

Reviewer #2: Yes

2. Has the statistical analysis been performed appropriately and rigorously?

Reviewer #1: Yes

Reviewer #2: Yes

3. Have the authors made all data underlying the findings in their manuscript fully available?

Reviewer #1: No

Reviewer #2: Yes

4. Is the manuscript presented in an intelligible fashion and written in standard English?

Reviewer #1: Yes

Reviewer #2: Yes

5. Review Comments to the Author

Reviewer #1: Kawakami et al. utilized dog samples and photographs that were submitted to Embark for commercial genetic profiling. The authors used the data to investigate the genetic basis of several canine coat color phenotypes, roaning, ticking, and flecking. The authors performed GWAS and identified an association signal for roaning on chromosome 38. Further analyses revealed an intronic 11 kb duplication in the USH2A gene encoding usherin, which is very strongly associated with the roaning phenotype. Somewhat surprisingly, the authors did not detect any association signal for ticking. The analysis on flecking is very indirect and circumstantial. It is based on the hypothesis that flecking in Dalmatians is linked to the hyperuricosuria locus, which was published in 1976 (Schaible et al.), before the availability of genetic markers. I was not able to access the full text of this 1976 publication. Given that the authors also show that the previous assumption on an interaction between ticking and flecking in Dalmatian spotting is apparently not true, I consider the work on flecking as too superficial and premature to be published at this time. The main result of the study is the identification of a plausible candidate cuasative variant for roaning. Demonstrating a causal role of USH2A for this phenotype would be an important finding of broad scientific interest. Unfortunately, the authors only report in (too) great detail about the identification of the variant, but provide no functional follow-up or at least a plausible mechanistic hypothesis how the duplication in USH2A might exert its effect.

Major comments:

(1) Some functional confirmation experiments are essential. Is there a qualitative (splicing) or quantitative effect on USH2A splicing? This should not be too difficult to assess in heterozygous animals where the transcripts from the wildtype and mutant allele can be directly compared to each other (e.g. by RNA-seq).

AU: We agree that a transcriptional study can provide an additional piece of information toward the better understanding of a relationship between genotypes and phenotypes. Spatio-temporal variation of gene expression can be particularly important to understand the genetic mechanisms of roaning because this phenotype (and ticking and Dalmatian’s spots as well) starts developing when a dog is around 2-6 weeks old. In order to unambiguously detect splicing variants and possible expression differences, one needs to carefully design a study to collect appropriate skin biopsy samples from tissues with or without roaning at several developmental stages (from newborn to adult). While this is a valid experiment, we believe that this is beyond the scope of this study where we aimed to identify a genomic region associated with roaning.

(2) At least a hypothesis should be presented how a dominant USH2A allele (gain of function?) might lead to increased pigmentation in the "white" patches of white spotted dogs.

AU: USH2A and MITF are involved in cell adhesion and endothelial cell migration, including pigmentation cells, so the interaction of regulatory mutations in both genes leading to novel pigmentation patterns is certainly plausible. As discussed earlier, transcriptional validation looking at transcript abundance and alternative splicing in relevant tissues (skin samples at the very least) during the proper developmental time period (likely early post-natal given the appearance of roaning marks, or possibly earlier if the relevant gene expression only occurs during cell migration) in dogs with and without the USH2A and MITF mutations would be required to develop a mechanistic understanding of the interaction of these genes in the presences of these mutations. Without functional experiments showing cellular localization of USH2A protein by using GFP, it is difficult to generate a realistic hypothesis indicating how this gene may interact with MITF and other pigmentation-related proteins. Given the limited understanding of these processes (despite the human medical importance of these genes), it is speculative to try to infer a possible functional mechanism, other than to say that these are clearly regulatory variants and not loss/gain-of-function (which also may explain the variable expression of both white spotting and roaning)." We are, of course, open to any suggestions if the reviewer may offer a hypothesis to add in the following paragraph. 

Line 707-727:

“This left the 11-kb tandem duplication as the strongest candidate for roaning, but follow-up functional validation is needed to investigate the exact impact of this duplication on the development of pigmented fur in otherwise unpigmented area. Our comparative analysis identified putative regulatory regions within and nearby the duplication that are highly conserved in vertebrates (S12 Fig). USH2A encodes the protein usherin. In humans, USH2A mutations are associated with Usher syndrome, characterized by progressive hearing loss and vision impairment often accompanied by retinitis pigmentosa [31]. A functional assay by using USH2A knockout mice showed that this gene is involved in the maintenance of retinal photoreceptors and the development of cochlear (inner ear) hair cells [32]. A recent study showed that a mutation in USH2A showed abnormal pigment deposition and reduced expression of MITF and other melanin metabolism-related genes, such as tyrosinase (TYR) and oculocutaneous albinism II (OCA2) genes, in retinal cells derived from induced pluripotent stem cells, indicating a potential involvement of USH2A in the pigmentation pathway [33]. Interestingly, the distribution of usherin in healthy individuals is highly conserved between mice and humans, in which skin was completely devoid of this protein [34,35]. We speculate that the duplication of the putative regulatory regions may result in ectopic expression of USH2A in skin melanocytes. Alternatively, the duplication may facilitate alternative splicing and create a novel protein isoform since this complex gene with 73 exons is known to form several isoforms [31,36]. High prevalence of congenital deafness in Dalmatians, Australian Cattle Dogs, and other commonly roaned breeds [37] may also imply a potential pleiotropic effect of USH2A on both hearing and pigmentation [38].”

(3)

The manuscript gives a huge amount of details on various imputation procedures, haplotype phasing, and other means to extract maximum information from the existing SNP chip genotypes. While this is admittedly very important for a diagnostic lab, it yields little insight into the biology of the trait. I wonder whether the manuscript could be restructured in a way that the manuscript also becomes informative and interesting to readers with a more focussed interest in pigmentation biology. Could some of the technical/diagnostic details be moved to supplementary data (perhaps as a supplemenatry methods text)?

AU: We moved methods and results describing genotype imputation and the association between the imputed genotypes and phenotypes (roaning or Dalmatian’s spot) to S1 Text.

(4)

Lines 380-382: "Since Variant Effect Predictor (VEP) [25] suggested that none of these variants had a large impact on the function of USH2A...". This statement is not true. VEP predicted several missense and frameshift variants. These were only excluded due to their genotype distribution, not due to the VEP prediction. The presentation of the manuscript makes it very difficult to follow the argumentation of the authors. The chapter "functional annotation" should be placed here (~line 380), before the search for structural variants.

AU: We corrected the sentence in Lines 380-382 as follows:

“Since Variant Effect Predictor (VEP) [25] suggested that none of the moderate-to-high impact variants were perfectly associated with roaning…” (Line 397-398). In addition, the sentences in the subsection "functional annotation" were moved in the paragraph starting with “There were 4,569 previously known single nucleotide variants (SNVs) and small indels…” in Line 380.

(5)

The work on flecking is too premature and inconclusive. The authors either need to present more complete data including a plausible mechanistic hypothesis how the flecking allele could work together with the roaning allele to produce the large solid spots in Dalmations or the entire sections should be reduced to a very short statement that the data on flecking are inconclusive (similar to the situation with ticking).

AU: Following the reviewer’s and the editor’s suggestions, we reduced the discussion of flecking. Removed sentences are following:

Line 740-:

“Three loci, namely F-locus, S-locus, and T-locus, have been proposed to be involved in the formation of distinctive spots in Dalmatians [39]. S-locus has been molecularly characterized, and the sw variant at MITF is required to have white fur as a base color (i.e., extreme white) [20,21]. Similar to other breeds with ticking, T-locus has been proposed to be a responsible locus for creating “ticks” or pigmented spots to the white coat [30] but, with a modifier F-locus on CFA3, causing fewer and larger spots without intermingling with the white fur in the base coat [8,22]. F-locus is responsible for “flecking” [39] but has not been molecularly characterised.”

Line 757-:

“The marker at CFA3:72,316,930 may represent one of the candidates linked with a causal variant of F-locus, although the genotypes of this marker were not perfectly associated with Dalmatian-like spots (S10 Table). Since flecking is defined as unpigmented fur within a pigmented region, phenotype characterization by photo examination may not be sufficient to correctly identify flecking. In addition, our F-locus search was agnostic about the genotypes of T-locus which may be one of the interacting loci with F-locus and the CFA38 duplication.”

Line 769-:

“We hypothesize that decoupling the allelic combinations at the modifier locus on CFA3 and the roaning locus on CFA38 reveals the putatively ancestral roaning coat pattern.”

(6)

File S1: PLoS journals require deposition of the complete raw data. The file with the marker names (bim-file) is insufficient. The authors should make bed-, bim-, and fam-files available or alternatively a ped and map file.

AU: We have now deposited all the raw data in Dryad, which will be freely available after satisfactory revisions (doi:10.5061/dryad.qz612jmd).

Minor comments:

(7)

Line 81: Roan in horses is linked to the KIT locus, not the KITLG locus.

AU: Horses are removed from the sentence.

(8)

Line 207: Insert an "and" between roaned and 567.

AU: Corrected.

(9)

Line 541: 120 roaned and 154 control dogs

AU: Corrected.

(10)

Line 618: It does not make sense to give amino acid exchanges for chromosomal (genomic) positions. The authors should give the amino acid variants with respect to specific proteins. HGVS nomenclature rules should be followed (https://varnomen.hgvs.org/).

AU: Corrected.

(11)

Figure S8: It should be explicitly stated what the marker in the magenta circle is. Is this the 11 kb duplication?

AU: Added a statement.

“The most significant CWAS marker is in magenta circle.”

(12)

Table S7: This table either needs much more detailed explanation or it can be deleted entirely. How do the authors determine K^B vs K^y when it is unknwon whether any K^br alleles are present? This will be difficult with an illumina array. To the best of my knowledge there are at least 3 different published e-alleles and 5 different b-alleles. The causal variant underlying the various agouti alleles ahve not been published (or under dispute). If this table shall remain in the manuscript, it must be explicitly stated which markers were genotyped to predict the coat color genotypes.

AU: S7 Table is removed.

(13)

File S9: This is a relatively small table. This should be given as a Supplementary table (Excel-file), rather than as a compressed zip-file.

AU: We wish to provide this table along with other supplementary files as all of these are raw analytical outputs. 

Reviewer #2: The authors use citizen science provided by dog owners in the form of photos of their dogs to find molecular explanation for coat color patterning in dogs. This is a lengthy study and authors mostly do a great job in describing this study in detail. The manuscript is technically sound, and data support the conclusions. The authors have made data underlining the data adequately available. The manuscript is written in standard English.

Minor comments:

1) The authors have had various breeds in the non-roan control group that should be mentioned and instead not include at all the breed Labrador Retriever that would be phenotypically always non-roan despite of the roaning genotype it might have due to this breed not presenting any color patches of white were the roan could be observed.

AU: A sentence was added in line 159.

“Breeds that never or rarely exhibit white spotting patterns were not included in the discovery panel.”

2) Chromosome-wide association analysis (CWAS) is incorrectly abbreviated.

AU: Corrected (Line 483). Note that this sentence is now in S1 Text by following the reviewer 1’s comment.

6. PLOS authors have the option to publish the peer review history of their article (what does this mean?). If published, this will include your full peer review and any attached files.

Do you want your identity to be public for this peer review? For information about this choice, including consent withdrawal, please see our Privacy Policy.

Reviewer #1: No

Reviewer #2: No

---

## [Editor Report · Decision Letter 1]

23 Feb 2021

R-locus for roaned coat is associated with a tandem duplication in an intronic region of USH2A in dogs and also contributes to Dalmatian spotting

PONE-D-20-35246R1

Dear Dr. Kawakami,

We’re pleased to inform you that your manuscript has been judged scientifically suitable for publication and will be formally accepted for publication once it meets all outstanding technical requirements.

Kind regards,

Christian Braendle

Academic Editor

PLOS ONE
---

## [Editor Report · Acceptance letter]

15 Mar 2021

PONE-D-20-35246R1 

R-locus for roaned coat is associated with a tandem duplication in an intronic region of *USH2A* in dogs and also contributes to Dalmatian spotting 

Dear Dr. Kawakami:

I'm pleased to inform you that your manuscript has been deemed suitable for publication in PLOS ONE. Congratulations! Your manuscript is now with our production department. 

Kind regards, 

on behalf of

Dr. Christian Braendle 

Academic Editor

PLOS ONE